# Trigger factor both holds and folds its client proteins

Kevin Wu [1], Thomas C. Minshull[2], Sheena E. Radford [2], Antonio N. Calabrese [2✉] &
James C. A. Bardwell [1✉]

ATP-independent chaperones like trigger factor are generally assumed to play passive roles in protein folding by acting as holding chaperones. Here we show that trigger factor plays a more active role. Consistent with a role as an aggregation inhibiting chaperone, we find that trigger factor rapidly binds to partially folded glyceraldehyde 3-phosphate dehydrogenase (GAPDH) and prevents it from non-productive self-association by shielding oligomeric interfaces. In the traditional view of holding chaperone action, trigger factor would then be expected to transfer its client to a chaperone foldase system for complete folding. Unexpectedly, we noticed that GAPDH folds into a monomeric but otherwise rather native-like intermediate state while trigger factor-bound. Upon release from trigger factor, the mostly folded monomeric GAPDH rapidly self-associates into its native tetramer and acquires enzymatic activity without needing additional folding factors. The mechanism we propose here for trigger factor bridges the holding and folding activities of chaperone function.

[1] Department of Molecular, Cellular, and Developmental Biology and Howard Hughes Medical Institute, University of Michigan, Ann Arbor, MI, USA. [2] Astbury Centre for Structural Molecular Biology, School of Molecular and Cellular Biology, Faculty of Biological Sciences, University of Leeds, Leeds LS2 9JT, UK. ✉email: A.Calabrese@leeds.ac.uk; jbardwel@umich.edu

Most proteins need to fold into a well-defined three-dimensional structure to perform their cellular function. While small single-domain proteins often can fold spontaneously and on physiologically relevant time scales, large multi-domain proteins usually require the assistance of molecular chaperones to prevent protein aggregation and facilitate protein folding[1]. Molecular chaperones are commonly classified into two categories based on their energy dependence. Chaperones that utilize ATP as an energy source can be considered to be "foldase" chaperones based on the discovery that they actively promote protein folding using the energy of ATP binding/hydrolysis to coordinate client binding and release[2,3]. In addition to foldase activity, evidence suggests that certain ATP-dependent chaperones can catalytically unfold misfolded client proteins, thus accelerating their productive refolding reactions[4–8]. Another class of chaperones are ATP-independent, which are commonly regarded as "holding" chaperones. It is usually assumed that the main function of these ATP-independent chaperones is to prevent protein aggregation by tightly holding onto and sequestering their client proteins in a non-native state. They are not thought to be directly involved in protein folding/unfolding processes[9,10]. This concept, however, has been challenged by emerging evidence showing that some chaperones can guide their client proteins to fold correctly in an ATP-independent fashion. An example is the *E. coli* chaperone Spy, which maintains protein folding homeostasis in the periplasm under stress conditions[11]. At least for some clients, Spy does not appear to be a holding chaperone. Instead, it loosely associates with its client proteins and allows them to fold while they remain chaperone bound[12,13]. We have recently reviewed the evidence that several other ATP-independent chaperones may also use a similar mechanism to assist in protein folding[14].

Trigger factor transiently engages with most newly synthesized polypeptide chains as they emerge from the ribosome[15]. As a bacterial ATP-independent chaperone, the trigger factor was initially thought to play only a passive role in protein folding, functioning by preventing protein aggregation and degradation[16]. The trigger factor is assumed to work by transferring nascent chains to ATP-dependent chaperone systems, such as the DnaKJE system and GroEL, for subsequent folding assistance[17–19]. While this is one possible fate for proteins bound by trigger factors, there is evidence that trigger factors can participate in protein folding in a more active way. For instance, the trigger factor has been found to facilitate the folding of maltose-binding protein by stabilizing on-pathway intermediate states[20]. Somewhat surprisingly, the trigger factor has also been shown to unfold pre-existing misfolded proteins, although this unfoldase activity is highly dependent on the thermodynamic stability of the client protein[21]. While evidence for the foldase activity of the trigger factor has been reported[20], the detailed structural and kinetic mechanism whereby the trigger factor associates with a refolding client protein and mediates productive protein folding remain largely unknown.

Previous studies demonstrated that trigger factor associates with short segments of polypeptides with affinities in the hundred micromolar range[22]. This weak affinity is postulated to allow the trigger factor to interact promiscuously with many nascent polypeptides during translation[23], and is viewed as an important feature of the trigger factor's chaperone activity[15,24]. More recently, NMR studies have demonstrated that the trigger factor binds to a fully unfolded peptide at multiple sites on its surface[25], consistent with its holding activity preventing unfolded protein from aggregation. In contrast to using either peptides derived from full-length proteins or clients engineered to be trapped in an unfolded state[24,25], we chose instead to use full-length glyceraldehyde 3-phosphate dehydrogenase (GAPDH) as a model client protein for trigger factor. It has been previously shown that the

trigger factor assists GAPDH reactivation, although the detailed mechanism by which this is achieved has remained largely unknown[26,27]. Here, we demonstrate that trigger factors can effectively prevent GAPDH from non-productive oligomerization. The trigger factor does so by rapidly forming a stable complex with monomeric, partially folded GAPDH. Using chemical cross-linking mass spectrometry (XL-MS) and hydrogen–deuterium exchange mass spectrometry (HDX-MS), we show that the trigger factor specifically recognizes regions that largely overlap with the inter-subunit interfaces of natively tetrameric GAPDH, which in turn are predicted to largely overlap with regions that are capable of driving aggregation. This chaperone–client binding mode effectively outcompetes non-productive intermolecular interactions within GAPDH molecules, thereby inhibiting mis-assembly/aggregation. Furthermore, we provide evidence demonstrating that the trigger factor not only holds onto non-native GAPDH to prevent aggregation but also allows it to fold into a structurally folded near-native state. Once released from the trigger factor, this assembly-competent intermediate state of GAPDH can self-associate into an active, tetrameric state. We additionally show that a similar mechanistic principle that we propose here for GAPDH folding/assembly can also be applied to another oligomeric client protein, namely *Vibrio harveyi* luciferase. These results highlight that an ATP-independent chaperone can bind client proteins to protect them from aggregation and permit folding while bound, and suggest that the trigger factor, as an ATP-independent chaperone, employs a hybrid holding–folding mechanism of chaperone action.

## Results

**Trigger factor rapidly forms a stable 1:1 complex with monomeric GAPDH.** One of the challenges in studying chaperone-mediated folding is the tendency of chaperone clients to undergo aggregation reactions. Although this helps define proteins as chaperone clients, aggregation also interferes with many types of biophysical measurements. The use of soluble peptides that bind to chaperones is one way around this issue but is not without its disadvantages. Peptides often have limited potential to gain secondary or tertiary structure, so peptide-based studies are more useful in exploring binding rather than folding reactions. Even when peptides are used to study binding, they often interact with proteins much more weakly than do the full-length, intact proteins from which they are derived, complicating mechanistic understanding. For instance, in solution, a trigger factor exists in a monomer–dimer equilibrium with an apparent $K_d$ of ca. 2–18 μM[28,29]. Given that the physiological concentration of trigger factor is ~50 μM[15], the majority of trigger factor molecules should be in a dimeric form. The NMR structure of the dimeric trigger factor forms a symmetric head-to-tail dimer, in which most of the known client binding sites are buried within the dimeric interface[28,30]. These structural data strongly suggest that the dimeric trigger factor would need to monomerize prior to association with its client proteins. Considering that most peptides that have been tested only exhibit weak binding affinity to trigger factor with $K_d$ values ranging from ~10 to 200 μM[22,24,25], it is unclear how a dimeric trigger factor would be effectively monomerized in the presence of client proteins if they bind less tightly to trigger factor than trigger factor binds to itself.

The identification of more physiologically meaningful chaperone–client pairs and conditions in which all components remain soluble is vital to obtaining a detailed understanding of a chaperone-mediated folding process. To simulate the process by which a chaperone encounters an unfolded or partially folded client protein, the most common method is to chemically denature the client and then dilute it out of denaturant in the presence of the chaperone. One client previously used, rabbit

GAPDH, unfortunately rapidly forms visible aggregates at room temperature upon dilution out of GdnHCl[31]. Fortunately, it has previously been shown that at low temperatures, GAPDH diluted out of GdnHCl is stable for days in a partially-folded molten-globule state, where it retains the potential to fold into the native state[32]. Unlike peptide substrates previously used for studying trigger factor–client interactions[24,25], this cold-stable molten-globule state of GAPDH opens the opportunity to explore both binding and folding reactions to trigger factors in ways that more closely resemble their interaction in a cellular context.

To examine the interaction between the trigger factor and the molten-globule state of GAPDH, GdnHCl-denatured GAPDH was diluted into a GdnHCl-free refolding buffer containing various concentrations of trigger factor. The size of the resulting complexes was analysed by performing sedimentation velocity-analytical ultracentrifugation and size-exclusion chromatography. Upon dilution in the absence of trigger factor, denatured GAPDH rapidly forms multiple soluble oligomeric species with a wide range of sedimentation coefficients ranging from 2.5 to 8.7 S, which correspond to oligomers ranging from dimers through tetramers to higher oligomers (Fig. 1a). Upon dilution into refolding buffer in the presence of trigger factor, however, GAPDH forms a single homogeneous complex with an apparent molecular weight of ~83 kDa as measured by sedimentation velocity-analytical ultracentrifugation, which is similar in size to the predicted molecular weight of 84 kDa for a 1:1 GAPDH-trigger factor complex (Fig. 1b; Supplementary Fig. 1). These results are consistent with our size-exclusion chromatography experiments, where we show that the addition of an equal amount of trigger factor resolves multiple species of refolding GAPDH into a single trigger factor–GAPDH complex (Fig. 1c). Increasing the concentration of trigger factor does not increase the molecular mass of the chaperone–client complex (Fig. 1c), suggesting that the trigger factor–GAPDH 1:1 complex does not appear to further assemble into a 2:2 complex, as was observed for the trigger factor–ribosomal protein S7 complex[33]. Given that monomeric GAPDH is two-fold larger in molecular weight than S7 (36 vs. 18 kDa), it is plausible that the client binding cavity formed by two trigger factor molecules might be too small to accommodate a single GAPDH molecule (Supplementary Fig. 2). Note that the excess of trigger factor used in our experimental conditions appears to exist as a monomer that sediments with a coefficient of 3 S (Supplementary Fig. 1c). No dimeric trigger factor was observed in these experimental conditions possibly due to the residual amount of GdnHCl (0.26 M) left in the refolding solution after dilution. Indeed, the dissociation constant of the dimerization of the trigger factor in these conditions was determined to be $76 \pm 6\,\mu\text{M}$ (Supplementary Fig. 3a), which is about 4-fold higher than the $18.4 \pm 0.8\,\mu\text{M}$ of $K_d$ we measured for trigger factor dimerization in the GdnHCl-free buffer (Supplementary Fig. 3a). Considering the concentrations of trigger factor that we used in this study is in the low µM range in most experiments, the majority of trigger factor molecules should be in the monomeric state. To test if the residual amount of GdnHCl in our experimental conditions could affect the activity of the trigger factor, we conducted GAPDH refolding experiments where the residual amount of GdnHCl after dilution was reduced to 0.06 M GdnHCl and measured the chaperone activity of trigger factor. We found that the trigger factor behaves similarly under these conditions (Supplementary Fig. 3b, c), although the $K_d$ value measured for trigger factor–GAPDH in 0.06 M GdnHCl buffer is $29 \pm 4\,\mu\text{M}$, 3-fold lower in 0.26 M GdnHCl buffer (Supplementary Fig. 3a). Additionally, to test if results were dependent on the type of denaturant, we conducted the GAPDH refolding assay after denaturing the protein in 8 M urea. We found that the trigger factor has similar chaperone activity in 0.7 and 0.16 M residual

urea even though the $K_d$ value measured for the dimerization of the trigger factor varies somewhat under these conditions (Supplementary Fig. 3a, d, e). Our results clearly show that the residual amount of GdnHCl present in our experimental conditions does not affect the trigger factor's chaperone activity. Given that the trigger factor has at least a 60-fold stronger binding affinity (~0.3 µM of $K_d$) to this non-native client protein as compared to the dissociation constant (~18 µM) for the self-association of the trigger factor, we reasoned that the presence of the non-native client protein GAPDH could readily trigger the monomerization of trigger factor, allowing monomeric trigger factor to protect this non-native client protein from aggregation.

Next, we determined the binding affinity of the trigger factor to a refolding GAPDH, using fluorescence anisotropy, and found it to be $0.32 \pm 0.08\,\mu\text{M}$ (Fig. 1d), indicating a tight association between trigger factor and refolding GAPDH. Using stopped-flow anisotropy, we further examined how fast trigger factor associates with the refolding GAPDH in order to prevent non-productive inter-subunit interactions. Upon dilution out of GdnHCl, a rapid increase in anisotropy occurred within the deadtime of the instrument (~5 ms) (Supplementary Fig. 4a), consistent with previous observations that denatured GAPDH rapidly folds into a molten-globule state through hydrophobic collapse[32]. Next, trigger factor was mixed with refolding GAPDH to examine how trigger factor associates with GAPDH. If trigger factor associates with non-native GAPDH prior to the formation of molten-globule state, the anisotropy value extrapolated to time zero should decrease as the trigger factor concentration increases. Instead, the anisotropy value extracted to time zero did not change upon addition of trigger factor (Supplementary Fig. 4b), suggesting that trigger factor does not associate with an extended, GdnHCl-denatured state but with a more compact molten-globule state of GAPDH. In addition, the rapid change associated with the hydrophobic collapse of GAPDH was followed by a slower but still rapid change in fluorescence anisotropy (Supplementary Fig. 4a). The observed rate constant ($k_{obs}$) for this slower phase increased linearly with trigger factor concentration (Supplementary Fig. 4c), consistent with a single-step bimolecular mechanism. Fitting the plot of $k_{obs}$ vs. the concentration of trigger factor to a linear curve gives an association rate constant ($k_{on}$) of $6.6 \times 10^6\,\text{M}^{-1}\,\text{s}^{-1}$ and a dissociation rate constant ($k_{off}$) of $8.7\,\text{s}^{-1}$ (Supplementary Fig. 4c). Combined, our data suggest that trigger factor rapidly binds and forms a stable complex with monomeric GAPDH after the formation of molten-globule state.

**Tight binding of trigger factor inhibits GAPDH self-assembly but promotes the refolding of GAPDH.** In the absence of a chaperone such as trigger factor, GAPDH aggregates and fails to regain its enzymatic activity upon refolding at 25 °C[26,27]. Productive GAPDH refolding is only observed when trigger factor is present[26,27], suggesting that trigger factor may play a crucial role in the GAPDH refolding process. However, given that GAPDH also aggregates under these conditions, it is difficult to dissect whether the increase in GAPDH refolding yield is due to trigger factor's anti-aggregation activity, a refolding activity, or both. To eliminate the effect of aggregation on the refolding yield and to examine whether trigger factor can still facilitate GAPDH refolding in the absence of aggregation, we performed GAPDH refolding assays on ice, conditions that eliminate GAPDH aggregation[32]. In the absence of trigger factor, little GAPDH enzymatic activity was detected even after a 3 h incubation time (Fig. 2a). However, GAPDH incubated with trigger factor regained substantial enzymatic activity after 30 min and the refolding reaction reached saturation, where it regains ~35% refolding yield compared to the same amount of native GAPDH,

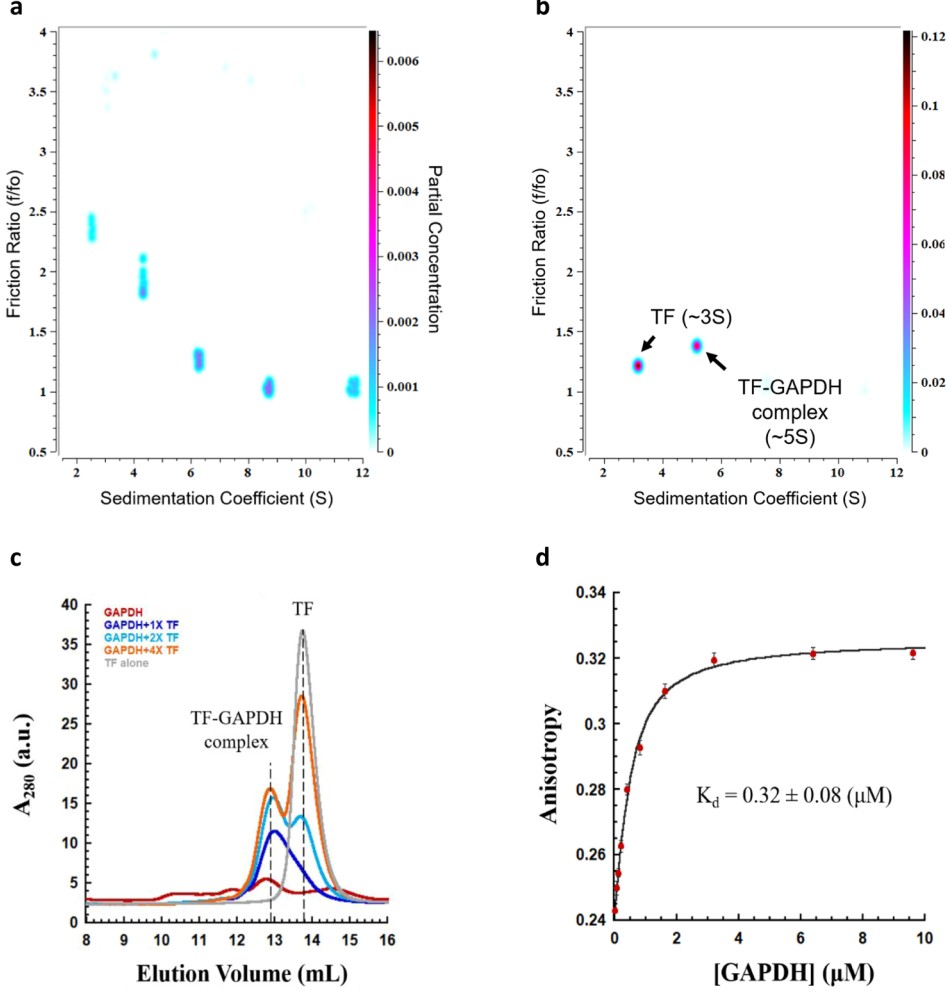

**Fig. 1 Interaction of trigger factor and refolding GAPDH. a** and **b** Sedimentation velocity-analytical ultracentrifugation analysis of **a** 5 µM refolding GAPDH and **b** 5 µM refolding GAPDH in the presence of 3-fold molar excess (15 µM) of trigger factor at 4 °C shows that the multiple species present for refolding GAPDH are resolved to a single TF–GAPDH complex in the presence of excess trigger factor. All samples were cooled down to 4 °C in the centrifuge over a period of 3 h, until the temperature equilibrated at 4 °C. All experiments were performed in buffer A containing 0.26 M GdnHCl, 4 °C, conditions under which the trigger factor is predominantly monomeric (Supplementary Fig. 3a). Data were analysed by two-dimensional sedimentation analysis, followed by a genetic algorithm-Monte Carlo analysis. **c** SEC profiles were obtained from 5 µM refolding GAPDH in the absence and in the presence of increasing concentrations of trigger factor (5–20 µM) at 4 °C. All samples were freshly prepared and immediately loaded onto a gel filtration column, which was pre-equilibrated with buffer A containing 0.26 M GdnHCl. a.u.: arbitrary units. **d** Fluorescence anisotropy binding curves of fluorescently labelling trigger factor upon titration with denatured GAPDH. The titration experiment was performed in buffer A at 4 °C. The data points were fitted to a quadratic equation to determine the dissociation constant ($K_d$). Each anisotropy measurement is the average of 20 independent measurements, and the error reported in the plot is the standard deviation. Value of $K_d$ reported is the mean ± s.e.m. of the fit.

after 90 min (Fig. 2b). This shows that trigger factor can directly promote the folding of GAPDH without requiring assistance from other chaperones. Next, we sought to test if the refolding yield of GAPDH could be further increased by the addition of ATP-dependent chaperones, e.g., DnaKJE, or if the ~35% refolding yield we observed is the maximal refolding yield that we could obtain under our experimental conditions. To test this, we incubated the refolding GAPDH with an excess trigger factor on ice for 3 h. Then, DnaKJE was added to the solution and the mixture was incubated at room temperature for a further 1 h prior to the measurement of enzymatic activity. We found that the addition of DnaKJE could further increase the refolding yield from 35% to 50% (Supplementary Fig. 5). These results indicated that a fraction of GAPDH is not able to fold or assemble correctly along the trigger factor-mediated folding pathway. Some of these GAPDH molecules may form a kinetically trapped intermediate state that requires ATP-dependent

chaperones to potentially unfold them and consequently promote their productive folding as has been previously seen with DnaK[4,5].

Interestingly, we observed a clear lag phase prior to the onset of enzyme activity acquisition (Fig. 2b). Given that the trigger factor forms a stable complex with GAPDH with a $K_d$ in the nM range (Fig. 1b, d), we wondered if binding of trigger factor may delay the reactivation of GAPDH, resulting in the observed lag phase. To test this idea, denatured GAPDH was diluted into a refolding buffer that contained various concentrations of trigger factor, and GAPDH enzyme activity was measured following a 3-h incubation. We reasoned that increasing concentrations of trigger factor should favour GAPDH binding to trigger factor and should disfavour GAPDH tetramerization, thus lengthening the lag phase. Indeed, the length of the lag phase significantly increased as trigger factor concentration is increased (Fig. 2c), indicating that the binding of the trigger factor inhibits GAPDH oligomerization. To approach

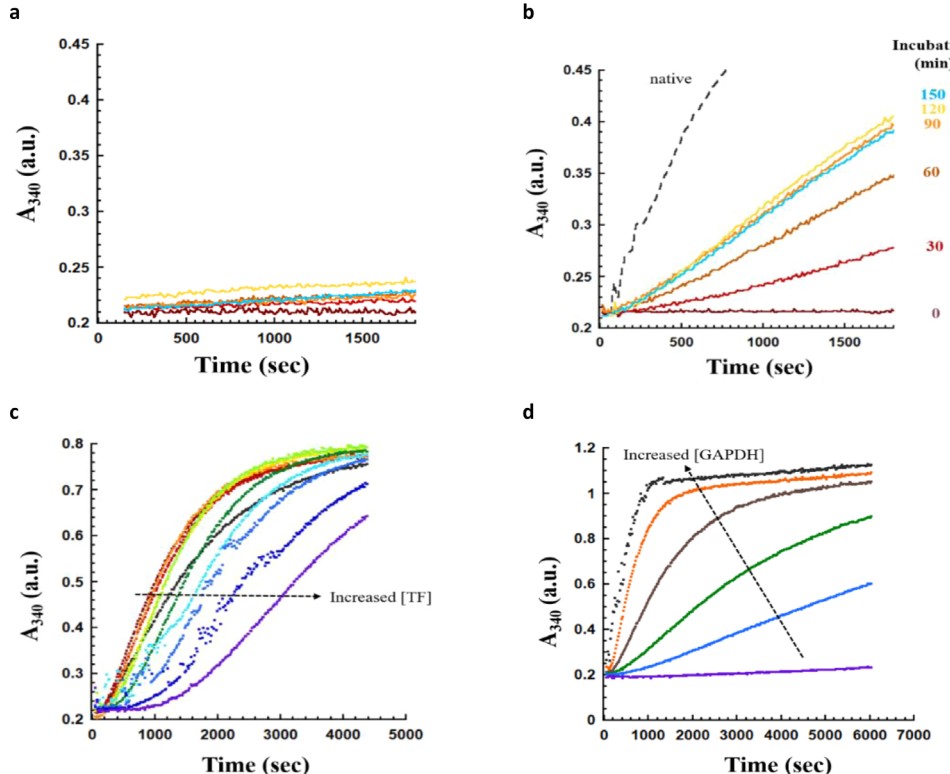

**Fig. 2 GAPDH regains its enzymatic activity in the presence of trigger factor. a** and **b** 2.78 μM refolding GAPDH was incubated on ice (**a**) in the absence of trigger factor, and **b** in the presence of equimolar amount of trigger factor. Very little activity is gained in the absence of trigger factor, but in the presence of trigger factor up to 35% of the original activity is recovered. All the samples were incubated on ice for the indicated time before 100-fold dilution into the activity buffer. The GAPDH activity was monitored by measuring the increase in absorbance at 340 nm. a.u.: arbitrary units. **c** 2.78 μM refolding GAPDH was incubated with various concentrations of trigger factor (5.6 to 89 μM) on ice for 3 h before measuring the GAPDH enzymatic activity. Increasing trigger factor concentrations lengthen the lag phase prior to the activity recovery. a.u.: arbitrary units. **d** Various concentrations of refolding GAPDH (0.35–11.12 μM) were mixed with trigger factor in a 1:2 molar ratio. The solutions were incubated on ice for 3 h before measuring the GAPDH enzymatic activity. Increasing GAPDH concentrations shorten the lag phase, suggesting that GAPDH forms a tetrameric, active, state upon release from the trigger factor. a.u. arbitrary units.

this from the opposite direction, with increasing concentrations of GAPDH, the tetramerization reaction should accelerate and the lag phase should be shortened. Consistent with this model, the lag phase was shortened upon increasing the concentration of GAPDH (Fig. 2d), suggesting that the trigger factor inhibits GAPDH assembly into an active tetrameric state.

**Trigger factor allows monomeric GAPDH to fold into a near-native state**. Next, we were curious as to how trigger factor facilitates GAPDH reactivation even though it inhibits GAPDH tetramerization (Fig. 2b), a reaction which is essential for GAPDH to achieve its enzymatically active native state[34,35]. One possibility is that though trigger factor inhibits GAPDH tetramerization, the chaperone can actively support the refolding of monomeric GAPDH. In this case, we should see structural differences between the unbound, molten-globule state of GAPDH and its trigger factor-bound state. To test whether molten-globular GAPDH undergoes conformational changes when bound to trigger factor, we monitored the tryptophan fluorescence spectrum of GAPDH in the absence and in the presence of trigger factor. GAPDH contains three tryptophan residues per monomer, located at positions 85, 194 and 311 in its primary sequence. The fluorescence of these tryptophan residues is a sensitive indicator of the folding status[32]. In order to examine the fluorescence signal of GAPDH in the absence of an interfering signal from trigger factor, we used a previously characterized

variant of trigger factor (W151F)[22], which has a similar GAPDH folding activity as the wild-type trigger factor does (Supplementary Fig. 6). Strikingly, the tryptophan fluorescence spectrum of trigger factor bound GAPDH is similar to the spectrum of native GAPDH, and distinct from the spectrum of refolded GAPDH and denatured GAPDH in terms of its intensity and the maximal emission wavelength (Fig. 3a). This strongly suggests that GAPDH folds from a molten-globule state to a near-native state in the presence of trigger factor. To examine how fast GAPDH folds into this near-native state, we monitored the change in the intensity of tryptophan fluorescence at 320 nm upon diluting the denatured GAPDH into the buffer containing the trigger factor W151F variant. In the absence of the W151F trigger factor variant, the initial tryptophan fluorescence is increased upon diluting out the GdnHCl, indicating that refolding GAPDH rapidly forms a molten-globule state with a fluorescence intensity higher than the GdnHCl-denatured state but lower than the native state (Fig. 3b). The formation of this molten-globule state is followed by a slow increase in the fluorescence intensity (Fig. 3b). Consistent with our stopped-flow binding kinetics data that trigger factor binds to refolding GAPDH after the formation of molten-globule state (Supplementary Fig. 4a), addition of trigger factor does not affect the initial tryptophan fluorescence of GAPDH (Fig. 3b). Importantly, we observed that the amplitude of the slow phase increases as the concentration of trigger factor W151F variant increases, approaching a similar fluorescence intensity as

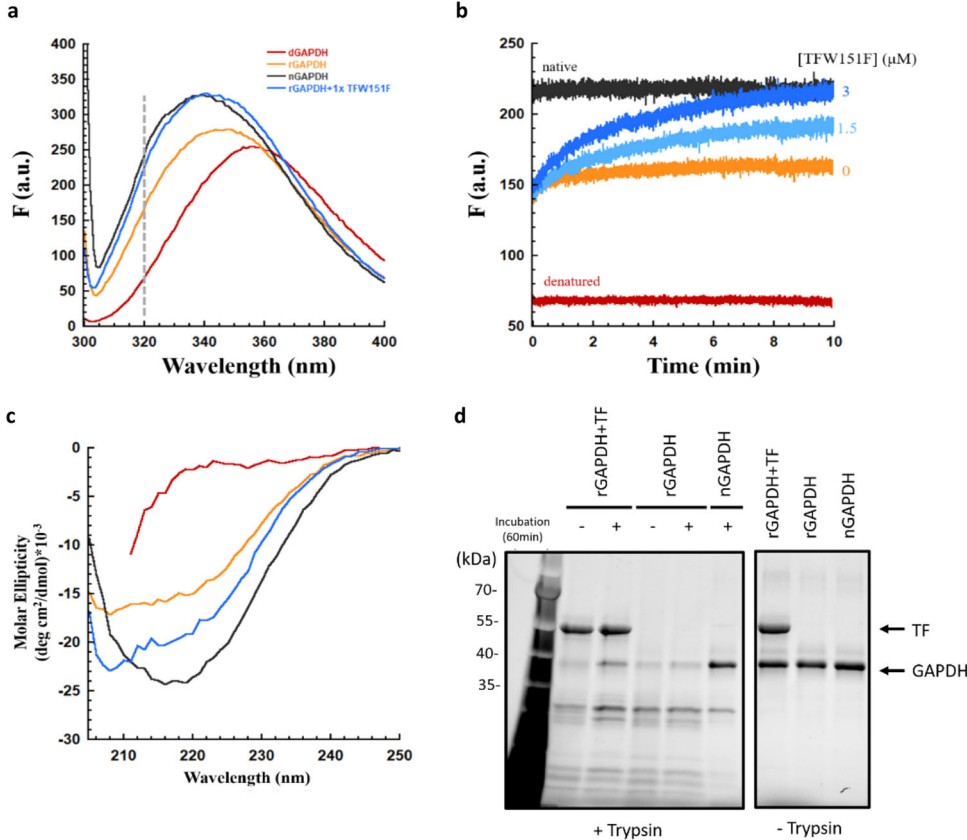

**Fig. 3 GAPDH slowly refolds to a near-native state in the presence of trigger factor. a** Tryptophan fluorescence spectrum of 3 μM refolding GAPDH (rGAPDH) with an equimolar amount of trigger factor W151F (rGAPDH+ 1x TFW151F; blue line) is similar to the spectrum of native GAPDH (nGAPDH; black line), but different from the one of GdnHCl-denatured GAPDH (dGAPDH; red line) and refolding GAPDH (rGAPDH; orange line). All the spectra were measured in buffer A containing 0.26 M GdnHCl at 4 °C, except the spectrum of GdnHCl-denatured GAPDH which was measured in buffer A containing 3 M GdnHCl. The emission wavelength of 320 nm, which was marked as a grey dash line, was used to monitor the kinetics in Fig. 3b. a.u.: arbitrary units. **b** Changes in tryptophan fluorescence of 3 μM refolding GAPDH in the presence of various concentrations of trigger factor W151F (0, 1.5, 3 μM) at 4 °C. Tryptophan residues were excited at 296 nm and fluorescence emission was monitored at 320 nm. Upon dilution into buffer A, GAPDH slowly refolds to a near-native state, which exhibited a similar fluorescence intensity as native GAPDH, within 10 min. a.u.: arbitrary units. **c** Circular dichroism spectra of 8.4 μM GAPDH in a denatured state (red line), native state (black line), refolding state (orange line) and the trigger factor-bound state (blue line). All the samples were incubated on ice for 1.5 h prior to recording the spectrum. Upon dilution into buffer A, a significant increase in the signal of circular dichroism at 220 nm was observed. Addition of the trigger factor further increased the signal at 220 nm, suggesting that GAPDH increases the fraction of secondary structure in the presence of the trigger factor. **d** The conformational change of 5 μM GAPDH in the presence of trigger factor as probed by limited proteolysis with trypsin. All the samples were either directly digested with 12.5 μg/ml trypsin for 1 min at room temperature or incubated on ice for 1 h before trypsin digestion. Samples were then analysed by 4–15% Mini-PROTEAN TGX stain-free gels. The gel shows that refolding GAPDH in the presence of trigger factor (rGAPDH + TF) is more protease-resistant than GAPDH in the absence of trigger factor (rGAPDH). These experiments were independently repeated three times with similar results.

native GAPDH (Fig. 3b). The data suggest that binding to trigger factor increases the population of a near-native, monomeric state of GAPDH, which is minimally populated in the spontaneous folding process. To achieve kinetic traces with better signal-to-noise, we rapidly diluted the GdnHCl-denatured GAPDH into the buffer containing various concentrations of trigger factor using a stopped-flow instrument (Supplementary Fig. 7a). Consistent with our observations in the fluorescence spectrophotometer, the amplitude of the fluorescence signal increases with increasing concentrations of trigger factor W151F, until it reaches a saturation value that occurs when the concentration of trigger factor is higher than the concentration of GAPDH (Supplementary Fig. 7b). Note that the observed rate constant ($k_{obs}$) for this observed slow change in tryptophan fluorescence with increasing concentrations of trigger factor could be fitted to a negative hyperbola, but it does not reach zero even at high concentrations of trigger factor (Supplementary Fig. 7c). Therefore,

even in circumstances where all molecules of GAPDH should be bound to trigger factor, a slow change in the conformation of the bound-form GAPDH still occurs. That folding continues under these circumstances strongly suggests that trigger factor allows monomeric GAPDH to fold to a near-native state while it remains chaperone-bound. To further interrogate the structural change of GAPDH while bound to trigger factor we next employed far-UV circular dichroism spectroscopy, which can inform on secondary structure content. The circular dichroism signal at 220 nm of the refolding GAPDH is increased upon addition of trigger factor, suggesting that the secondary structure content of GAPDH increased in the presence of trigger factor, compared to that observed in the absence of trigger factor (Fig. 3c). Moreover, we conducted limited proteolysis with trypsin and showed that refolding GAPDH in the presence of trigger factor becomes more protease-resistant after an hour of incubation (Fig. 3d). In contrast, refolding GAPDH in the absence of

trigger factor remains in a protease-sensitive state after an hour incubation (Fig. 3d). Thus, by using multiple approaches, we have demonstrated that trigger factor rapidly binds to the molten-globule state of GAPDH, whereupon it facilitates correct GAPDH folding by preventing non-productive interactions and by stabilizing GAPDH in a near-native, protease-resistant state, which is inaccessible in the absence of trigger factor.

These results are consistent with, and build upon, a previous study where the authors used maltose-binding protein as a model protein for folding[20]. Since maltose-binding protein is an extremely stable protein with a robust folding mechanism, truncations, and an artificial construct with four maltose-binding protein repeats were required to convert it into an aggregation-prone client. GAPDH, on the other hand, though prone to aggregation during its spontaneous folding processes, remains soluble when held at 4 °C in a molten globule partially oligomerized state. Using these conditions, we could directly test the effects of the trigger factors on the folding of a full-length client. The results show that the trigger factor maintains GAPDH in a monomeric form and enables it to fold to a near-native state while still bound to the chaperone. This species then rapidly tetramerizes and becomes enzymatically active following dissociation from the trigger factor. Our data could explain a previous in vitro observation that the refolding yield of GAPDH only increases over a certain range of trigger factor concentrations and decreases at higher concentrations of trigger factor[26,27]. This was initially attributed to the formation of a stable ~200-kDa complex of dimeric trigger factor and GAPDH at high concentrations[26,27]. However, using analytical ultracentrifugation and size exclusion analysis, we clearly showed that monomeric GAPDH is stabilized by monomeric trigger factor even at high concentrations of trigger factor, forming a ~80-kDa complex (Fig. 1b, c). Furthermore, we wondered if the decrease in refolding yield reported previously at high concentrations of trigger factor was also due to the lag phase. We thus decided to conduct the GAPDH refolding experiments at 25 °C as previously done[26,27]. Similar to what has been shown previously, the GAPDH refolding yield initially increases as trigger factor concentration increases and then decreases when trigger factor concentration further increases (Supplementary Fig. 8a). Note that the absorbance curve at high concentrations of trigger factor showed a clear curvature after 10 min (Supplementary Fig. 8b), during which GAPDH could fold into a native state upon release from trigger factor, which is much shorter than the time required for GAPDH refolding into a native state upon dilution of GdnHCl (>60 min)[26,27]. These observations are in line with our proposed mechanism that trigger factor enables GAPDH to fold into a near-native state while bound; upon release from trigger factor, this (partially) folded GAPDH can subsequently self-assemble into its active state rapidly.

**Trigger factor selectively shields oligomeric interfaces of GAPDH.** To investigate how trigger factor recognizes GAPDH, we used chemical crosslinking-mass spectrometry (XL-MS). XL-MS is a useful tool to inform on the topology of a protein or protein complex as it affords residue-level distance restraints that are of a length defined by the crosslinking reagent used[36,37]. First, we diluted 6 μM denatured GAPDH into a pre-chilled buffer containing an equimolar amount of trigger factor (where the final concentration of GdnHCl was 0.26 M). The mixture was then incubated on ice for 2 h to ensure that the majority of GAPDH molecules have folded into the near-native state upon forming a stable complex with trigger factor (Fig. 2b). Crosslinking was performed using the homobifunctional NHS-ester crosslinker disuccinimidyl dibutyric urea, which has been shown to react

with residues up to 30 Å apart (Cα–Cα Euclidean distance)[38]. A single band with a molecular weight on SDS–PAGE corresponding to a cross-linked 1:1 trigger factor–GAPDH complex was observed (Fig. 4a).

Following digestion and mass spectrometry analysis, 16 unique intermolecular crosslinked peptides between trigger factor and GAPDH were detected (Fig. 4b, Supplementary Table 1). Six out of the seven cross-linked residues on trigger factor were spread over the inner face of trigger factor's C-terminal domain (Fig. 4b, c). These results are consistent with previous analyses showing that trigger factor's C-terminal domain is essential for client recognition[25,39]. Only one cross-linked residue on trigger factor was found in the PPIase domain, which is in line with previous results that showed that it's PPIase domain is dispensable for trigger factor's chaperone activity both in vitro and in vivo[39]. Trigger factor's C-terminal domain comprises a "body" and two "arms" (arm 1 and arm 2), and together with the N-terminal domain is responsible for trigger factor's cradle-like structure (Fig. 4c). K327, a residue located at the tip of arm 1, was a crosslink hotspot, crosslinking with 7 out of the 9 residues in the client where crosslinks were obtained. This suggests that the tip of this arm is near the client and may be flexible. Several additional cross-linked sites (K272, K279, K287, and K342) are located throughout the body of the C-terminal domain, and one cross-linked site (K361) is on the second arm. Overall, the trigger factor client binding sites revealed here are similar to the client binding sites revealed by a previous NMR-based study using unfolded PhoA as a client (Supplementary Fig. 9a)[25]. One notable difference is an additional binding site, that was not identified previously, located in the bottom of trigger factor's cradle (residues 275–290), a region which is more hydro*philic* than other inner regions of the cradle (Supplementary Fig. 9b). In the NMR structural models, PhoA peptides are fully unfolded both in solution and when bound to trigger factor, which may explain why they interact with trigger factor mainly through hydrophobic interactions and do not utilize the hydrophilic client-binding region in the bottom of cradle[25]. By contrast with these data, the results presented here suggest that binding to a partially folded protein, like GAPDH, relies on both hydrophilic and hydrophobic residues exposed on the surface of trigger factor.

On the side of GAPDH, six unique lysine residues (K115, K192, K217, K249, K257 and K261) were found to crosslink with trigger factor (Fig. 4d). Five of these cross-linking sites (K192, K217, K249, K257 and K261) are located within GAPDH's catalytic domain which spans residues 150–314, and one cross-linked site (K115) is found in the cofactor-binding domain which spans residues 1–149; 315–333 (Fig. 4d). The oligomeric interface in native tetrameric GAPDH is extensive, spanning much of the catalytic domain consisting of a β-sheet and an extended loop (Fig. 4d). One cross-link (K192) was found in this extended loop and many others were found in the area surrounding the β-sheet (K249, K257, K261) (Fig. 4d), however no cross-links were found with K307, which is the only available lysine within the β-sheet itself. Although the area on GAPDH that binds trigger factor is not identical to the GAPDH tetramerization interface, there is extensive overlap between these sites. Our XL-MS experiments therefore suggest that binding of trigger factor to the catalytic domain of GAPDH could at least partially shield the enzyme from self-oligomerization and aggregation/native state formation.

**Conformational changes in GAPDH upon binding of trigger factor monitored by HDX-MS.** Next, to further investigate the effects of trigger factor binding on the conformational properties of GAPDH, we used differential hydrogen-deuterium exchange mass spectrometry (HDX-MS). HDX-MS informs on solvent

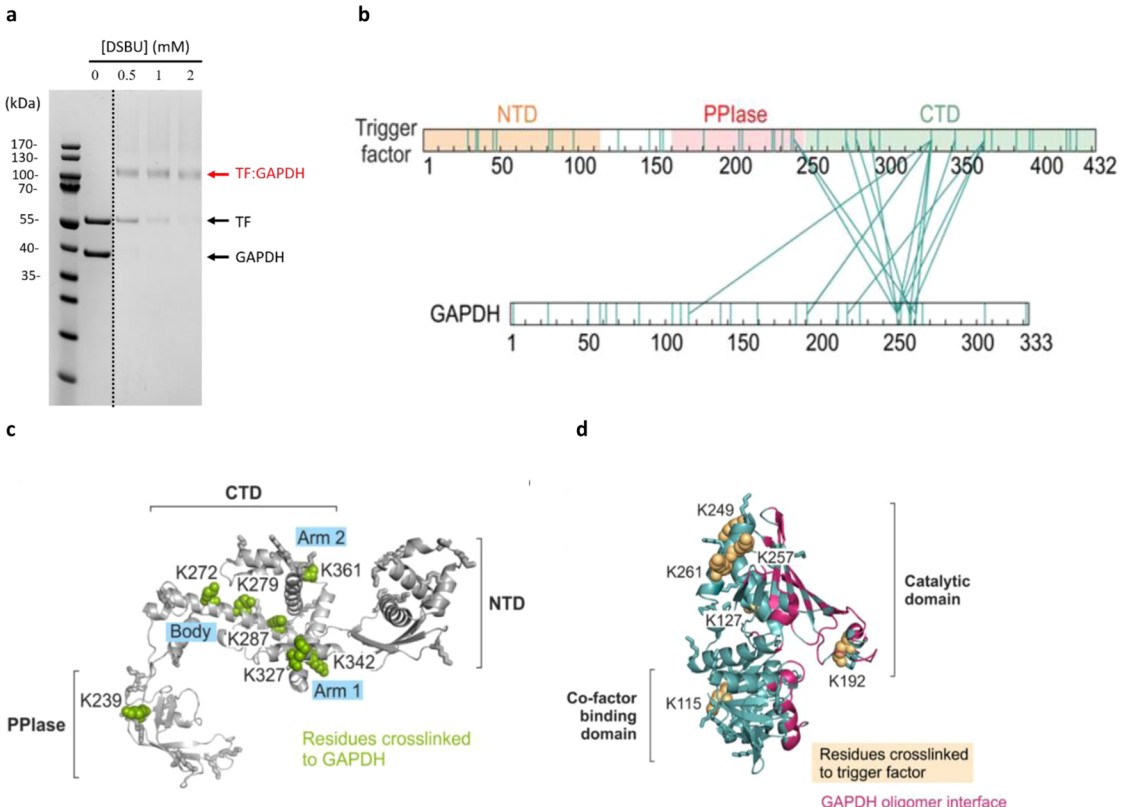

**Fig. 4 XL-MS reveals the binding interface of trigger factor–GAPDH complex. a** SDS–PAGE analysis of the disuccinimidyl dibutyric urea cross-linked species shows a homogeneous cross-linked trigger factor–GAPDH complex. Note that above the band corresponding to the crosslinked complex there are additional crosslinked species corresponding to higher order oligomeric assemblies, but these represent only a small proportion of the crosslinked products. These experiments were independently repeated three times with similar results. **b** The intermolecular cross-links found in the trigger factor–GAPDH complex are shown as teal solid lines. All the lysine residues across both protein sequences are marked in teal. Three structural domains of trigger factor, including the N-terminal domain (NTD), Peptidyl-prolyl isomerase domain (PPIase domain) and the C-terminal domain (CTD), are labelled. A list of all intermolecular crosslinked peptides can be found in Supplementary Table 1. **c** The cross-linked sites, shown as green spheres, are labelled and mapped onto the crystal structure of trigger factor (pdb: 1w26). Non-cross-linked lysine residues are shown as sticks. The three structural domains of trigger factor are labelled as outlined in **b**. **d** The cross-linked sites, shown as pale-yellow spheres, are labelled and mapped onto a subunit of crystal structure of rabbit-muscle GAPDH (pdb: 1J0X). The oligomeric interface of native GAPDH is coloured in pink. The two structural domains of GAPDH, catalytic domain and cofactor-binding domain, are labelled. Non-cross-linked lysine residues are shown as sticks.

accessibility and/or dynamics of the main-chain amide NH of proteins and has been shown to be a powerful technique to study chaperone–client interactions[40]. We first analysed the differences in deuterium uptake between trigger factor alone and trigger factor bound to GAPDH. Under the conditions used for these experiments (8 µM trigger factor, 8 µM GAPDH, final GdnHCl concentration of 0.26 M) trigger factor is predominantly monomeric (Supplementary Fig. 1c), and dilution from denaturant at low temperature favours the formation of the molten-globule state of GAPDH. We found that many regions across the inner face of the C-terminal domain of trigger factor are protected from deuterium exchange upon binding GAPDH (Fig. 5a, b), consistent with the XL-MS data presented above that implicates this surface in binding. We did not detect any peptides spanning residues 266–296 (Fig. 5b), which we identified as a client binding site by XL-MS (Fig. 4c), so could not obtain any information about this region from the HDX-MS experiments. However, we did observe protection from exchange in regions which are derived from residues near the putative binding site in this region identified by XL-MS (Fig. 5b). We also observed regions of protection in the N-terminal domain and the linker region that connects the N-terminal domain to the PPIase domain. Given that these regions have not been observed previously to be involved in client binding[39], and were not crosslinked to GAPDH

in our experiments, it is likely that this protection from exchange results from a change in the conformational dynamics of trigger factor upon binding. We also observed a region of deprotection in the PPIase domain of trigger factor upon GAPDH binding. This is consistent with data on the conformational dynamics of trigger factor, in particular molecular dynamics simulations which suggest the PPIase domain contacts the C-terminal domain transiently in the apo state[41]. This deprotection could be explained by an allosteric change in the conformation/dynamics of trigger factor upon binding GAPDH, potentially because of the PPIase domain undocking from the C-terminal domain.

Next, we compared the differences in deuterium exchange between refolding GAPDH (GAPDH diluted from denaturant in the absence of chaperone) and GAPDH bound to trigger factor. Surprisingly, we found that many regions mapping across the entire GAPDH structure are deprotected upon binding, becoming more solvent-exposed/dynamic in the bound state (Fig. 5c, d). This observation can be rationalised by the fact that refolding GAPDH tends to form oligomers in the absence of trigger factor (Fig. 1a). If we were comparing the extent of deuterium exchange in monomeric GAPDH in the bound and unbound forms, it would be expected that the regions of GAPDH involved in the trigger factor binding interface(s) would be protected from exchange in the complex. However, refolding GAPDH forms

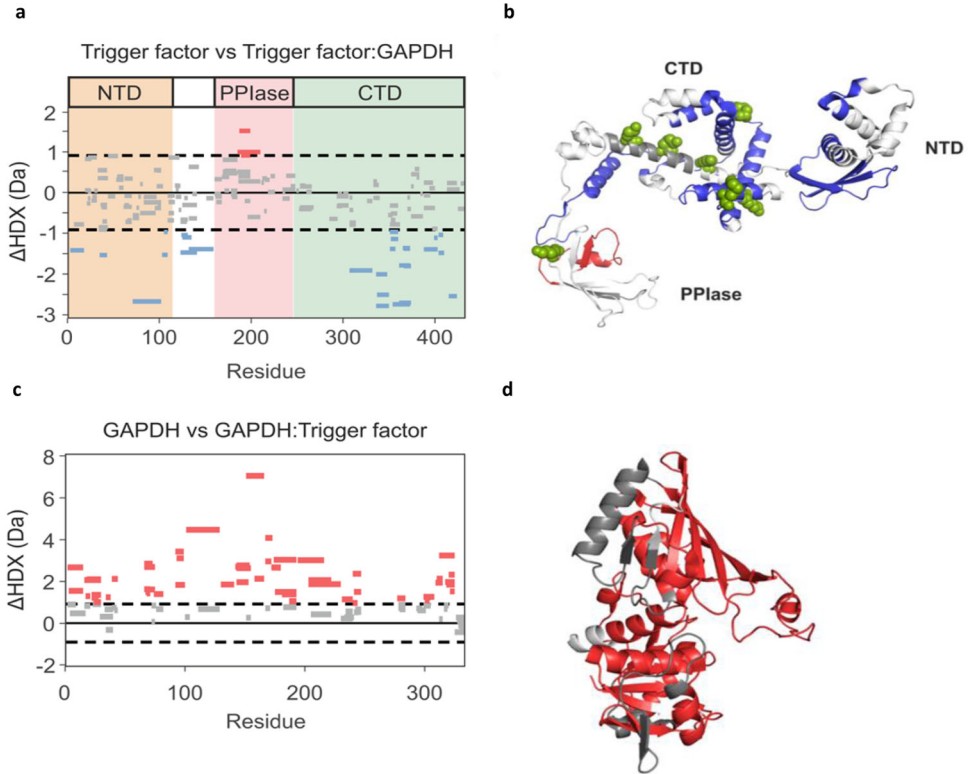

**Fig. 5 HDX-MS reveals GAPDH binding sites on trigger factor and suggests that bound GAPDH is monomeric. a** Woods plot showing the summed difference in deuterium uptake in trigger factor over all measured time points, comparing trigger factor alone and trigger factor in the presence of GAPDH. Peptides coloured blue and red are protected and deprotected from exchange in the presence of GAPDH (see the "Methods" section). Peptides with no significant difference between conditions, determined using a 99% confidence interval (dotted line), are shown in grey. **b** The differences in deuterium uptake plotted on the structure of trigger factor (PDB: 1JOX). Blue and red regions are protected or deprotected from exchange, respectively, in the presence of GAPDH. **c** Woods plot showing the summed difference in deuterium uptake over all measured time points (0.5, 2, 5, 10 and 30 min) between refolding GAPDH and GAPDH in complex with TF. Peptides coloured red are deprotected from exchange in the presence of GAPDH (see the "Methods" section). No peptides show statistically significant protection in the presence of GAPDH. Peptides with no significant difference between conditions, determined using a 99% confidence interval (dotted line), are shown in grey. **d** The differences in deuterium uptake plotted on the structure of GAPDH (right) PDB: 1JOX. Red regions are deprotected from exchange when bound to trigger factor.

stable oligomers, whereas monomeric GAPDH binds trigger factor. Consequently, the observed deprotection from deuterium exchange in GAPDH upon binding trigger factor may be because the chaperone prevents the formation of oligomers (which are protected from exchange), and in the bound monomer the protein is less protected from exchange. This is consistent with previous observations about ATP-independent chaperones, whereby weak and transient interactions between chaperone and client are key to protein folding[13]. To determine if the regions that are deprotected upon trigger factor binding have a tendency towards self-association, we used AGGRESCAN[42] to predict aggregation-prone regions present within the GAPDH structure (Supplementary Fig. 10). All the predicted aggregation-prone regions and oligomerization interfaces overlap with the deprotected regions revealed by these HDX-MS experiments (Supplementary Fig. 10). Combined, these data suggest that trigger factor recognizes the GAPDH oligomerization interfaces and aggregation-prone regions through a heterogenous binding surface that contains a mixture of hydrophobic and hydrophilic residues.

**Preventing aggregation while promoting folding appears to be a general trigger factor action.** It has been reported previously that many trigger factor's physiological clients oligomerize or assemble into a larger complex[33]. We wondered if trigger factor could also actively promote the folding of other oligomeric

proteins potentially using a similar mechanism that we have proposed here for GAPDH. To test this, we investigated *Vibrio harveyi* luciferase as a client protein. *Vibrio harveyi* luciferase is a heterodimeric flavin monooxygenase, that oxidizes long-chain aliphatic aldehydes while emitting photons[43]. During spontaneous refolding, luciferase self-assembles into multiple non-productive oligomers (Fig. 6a), which is similar to what we see in the spontaneous refolding of GAPDH (Fig. 1a). In the presence of trigger factor, a stable 1:1 complex between trigger factor and the luciferase subunits forms, and non-productive inter-subunit interactions between the luciferase molecules are disfavoured (Fig. 6b). The binding affinity of trigger factor to refolding luciferase subunits can be determined with a $K_d$ value of 4.5 ± 0.4 μM (Fig. 6c). While this is weaker than the apparent $K_d$ we measured earlier for binding GAPDH (0.32 ± 0.08 μM in Fig. 1d), it is still tighter than the apparent $K_d$ for the dimerization of trigger factor (Supplementary Fig. 3a). Consistent with what we found for GAPDH, trigger factor not only passively prevents luciferase from non-productive oligomerization but also actively promotes luciferase refolding. Increasing concentrations of trigger factor significantly increase the reactivation yield of luciferase from ~20% to ~60% (Fig. 6d). These data are similar to previous in vivo observations that have shown that trigger factor delays the co-translational dimerization of *Vibrio harveyi* luciferase, and allows its dimerization to occur only after the dimeric interface of another luciferase subunit has fully emerged from the ribosome[44].

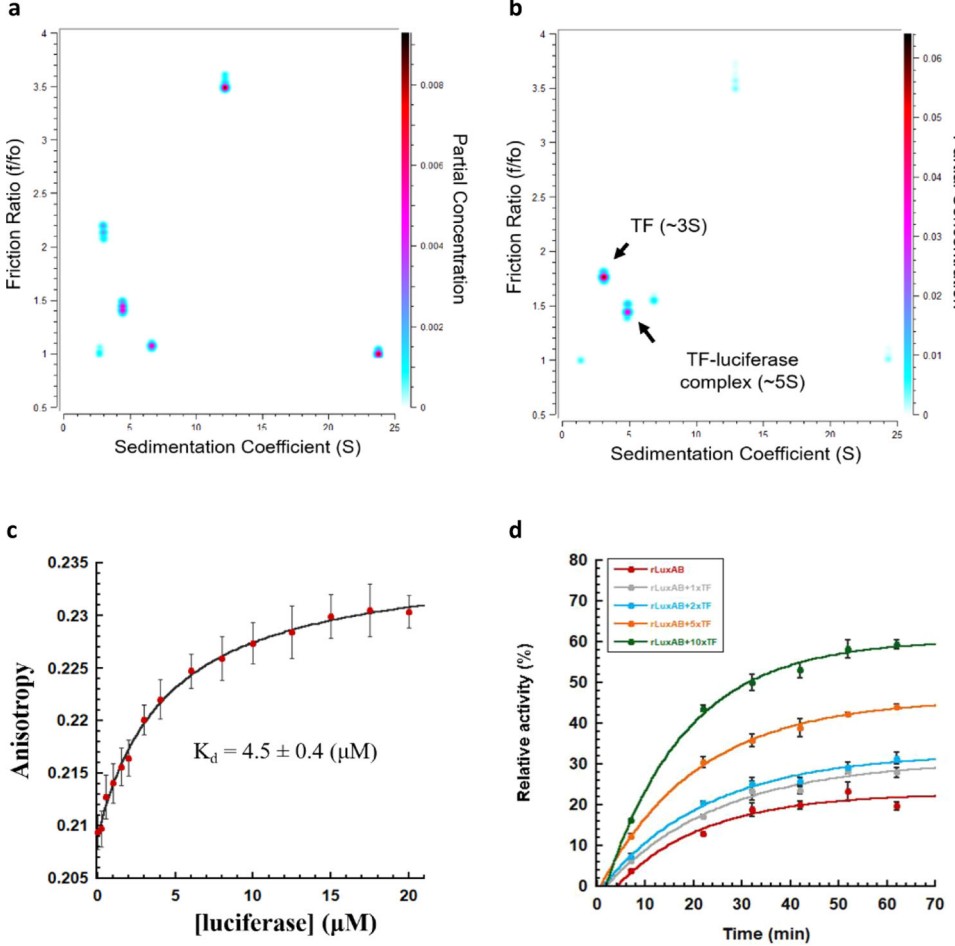

**Fig. 6 Trigger factor eliminates non-productive oligomerization and facilitates Vibrio harveyi luciferase refolding. a** and **b** Sedimentation velocity-analytical ultracentrifugation analysis of **a** 2 μM refolding luciferase and **b** 2 μM refolding luciferase in the presence of 8-fold molar excess (16 μM) of trigger factor at 4 °C shows that the multiple species present for refolding GAPDH are resolved to a single TF–GAPDH complex in the presence of excess trigger factor. All samples were cooled down to 4 °C in the centrifuge over a period of 3 h, until the temperature equilibrated at 4 °C. All experiments were performed in buffer A containing 0.1 M urea, 4 °C. Data were analysed by two-dimensional sedimentation analysis, followed by a genetic algorithm-Monte Carlo analysis. **c** Fluorescence anisotropy binding curves of fluorescently labelling trigger factor upon titration with urea-denatured luciferase. The titration experiment was performed in buffer A at 25 °C. The data points were fitted to a hyperbola equation to determine the dissociation constant ($K_d$). Each anisotropy measurement is the average of 20 independent measurements, and the error reported is the standard deviation. Value of $K_d$ reported is the mean ± s.e.m. of the fit. **d** In vitro bacterial luciferase activity assay. 1.06 μM refolding luciferase was incubated with various concentrations of trigger factor (0–10.6 μM) at 25 °C. The enzymatic activity of refolding luciferase was monitored at various refolding time points. Each data point is the average of three independent measurements, and the error reported is the standard deviation.

Our present in vitro data, together with these previous in vivo data, allows us to propose that trigger factor effectively prevents non-productive self-assembly by partially shielding the aggregation-prone oligomeric interface, in the meantime allowing its clients to explore folding until they reach a near-native state. Given that oligomeric interfaces are typically aggregation-prone[45–47], we believe that shielding oligomeric interfaces may be representative of a generic mechanism whereby trigger factor binds to otherwise unstable or aggregation-prone monomers so that they can be held in a folding competent state whilst waiting for their partners to arrive.

## Discussion

Our present study provides in-depth structural and mechanistic details of how trigger factor–client complexes form, and how trigger factor folds its client proteins. In summary, we propose a mechanism whereby trigger factor allows its clients to fold to an assembly-competent state while remaining bound to the

chaperone, protecting them from aggregation. Upon release from trigger factor, assembly competent, monomeric GAPDH subunits are primed to easily find and associate with their binding partners to form the native tetrameric state (Fig. 7). Given that a large portion of trigger factor's clients form high-order assemblies in their native, functional state[33], shielding potentially aggregation-prone oligomeric interfaces and preventing them from mis-assembly may be a general principle whereby trigger factor mediates co- and post-translational protein folding and assembly.

Our study also provides structural insights as to how trigger factor associates with an aggregation-prone partially folded protein. In contrast to the hydrophobic binding mode trigger factor has been shown to utilize for fully unfolded proteins[25], we find that trigger factor uses both hydrophobic and hydrophilic regions on its surface to interact with a partially folded protein. Interestingly, such a heterogeneous surface has also been observed in Spy's client binding sites and these different client binding modes have been shown to be an important feature for a client protein to

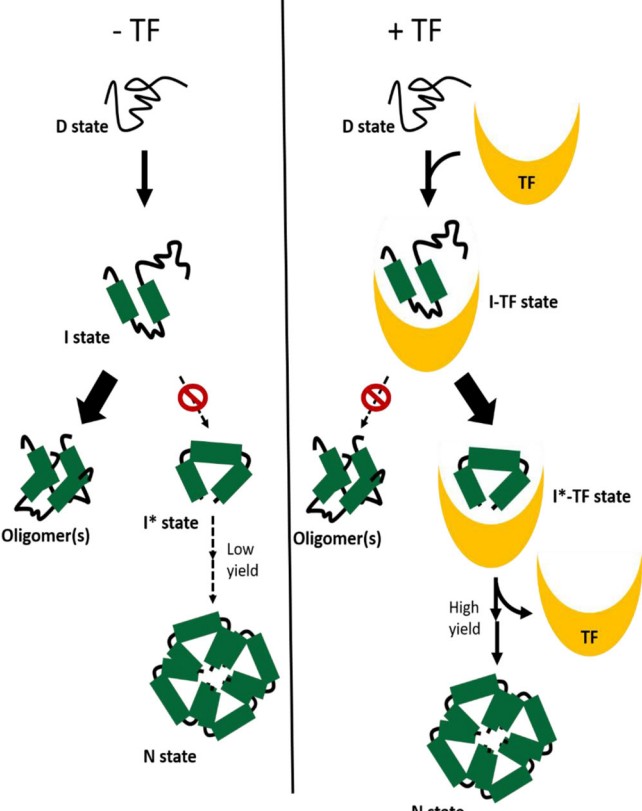

**Fig. 7 Model of trigger factor-mediated reactivation of oligomeric clients.**
Upon initiating refolding, denatured clients (D state) can spontaneously
form into a partially folded intermediate state (I state) through hydrophobic
collapse. In the absence of trigger factor (TF), the majority of GAPDH
molecules mis-assemble into non-productive oligomers, resulting a lower
refolding yield. By contrast, trigger factor shields these oligomeric
interfaces of partially folded intermediate state and prevents them from
aggregation. Trigger factor also allows these partially folded states to refold
into an aggregation-resistance intermediate state (I* state) while bound.
Once released from trigger factor, this folded client could self-assemble
into its active, oligomeric state (N state).

fold while bound to chaperones[11,48]. A similar switch from a
hydrophobic binding mode to a more hydrophilic binding mode
has been shown in the action of the chaperone GroEL[49]. GroEL
initially binds to unfolded proteins mainly through hydrophobic
interactions. Upon association with ATP and the co-chaperone
GroES, the interior of GroEL undergoes large conformational
changes and creates a more hydrophilic environment, which is
thought to favour protein folding[49]. However, unlike GroEL,
ATP-independent chaperones do not rely on ATP and co-
chaperones to regulate their chaperone activities. At least two
ATP-independent chaperones, trigger factor and Spy, appear to
use a semi-open binding pocket containing a mixture of hydro-
phobic and hydrophilic residues to associate with their client
proteins[11,23]. These heterogeneous client binding surfaces may
provide "folding-friendly" environments, wherein client proteins
can search for their native or near-native conformations as pro-
tein monomers while being protected from misfolding and
aggregation.

Structural information regarding the conformations of client
proteins bound to chaperones is a long-standing need in the
chaperone field. Fortunately, with the development of advanced
biophysical approaches, structural models of chaperone–client
complexes are now becoming available. These studies have shown
that many clients bound to ATP-independent chaperones are not
in a fully unfolded state, but rather adopt partially folded or even
near-native-like states[20,50–52]. Even small heat shock proteins,
which act as classic holding chaperones, have been shown
recently to stabilize their client proteins in a near-native
conformation[50]. Future work may reveal how these ATP-
independent chaperones can influence the conformations of
other client proteins, and how that, in turn, affects the folding fate
of these clients. But what is now clear is that rather than being
innocent bystanders of a folding reaction, ATP-independent
chaperones such as trigger factors can assist proteins to fold
towards their native states by allowing folding to occur while
bound. Accordingly, premature release of aggregation-prone
intermediates is avoided by the release of monomers that are
primed to be able to fold and assemble into their active states, as
exemplified here by two natively oligomeric enzymes, GAPDH
and luciferase.

## Methods

**Protein expression and purification.** Genes encoding wild-type trigger factor,
trigger factor variant W151F and E326C, DnaK, DnaJ and GrpE, were subcloned
into a pET28a vector containing an N-terminal His-Sumo tag and were trans-
formed individually into the *E. coli* BL21 (DE3) cell line (New England Biolabs).
Cultures were grown to an O.D.$_{600}$ of 1.0 at 37 °C and the expression of the protein
was induced overnight with 0.1 mM IPTG at 20 °C. Cell pellets were resuspended
in lysis buffer (50 mM Tris, pH8.0, 400 mM NaCl, 15 mM imidazole, 10% glycerol)
with EDTA-free Protease Inhibitor Cocktail (cOmplete™) tablet and 0.1 mg DNase
I. Samples were sonicated for 8 min on ice, then clarified by centrifuging twice at
36,000 × *g* for 30 min. The resulting supernatant was loaded onto a Ni-His Trap
column (GE Healthcare) equilibrated in lysis buffer. The column was washed with
lysis buffer, and the protein with a His-SUMO tag was eluted in lysis buffer with
500 mM imidazole. The eluate was dialysed into 50 mM Tris, pH 8.0, 300 mM
NaCl with ULP1 protease overnight at 4 °C, then the native protein was purified
away from the cleaved tag by passing over a Ni-His Trap column (GE Healthcare).
The untagged protein sample, now in the flow-through from the column, was
loaded onto a HiTrap Q column (GE Healthcare) equilibrated in 30 mM Tris, pH
8.5, 50 mM NaCl. The protein was then eluted using a NaCl gradient (50–400 mM)
in 30 mM Tris (pH 8.5). The remaining impurities were removed by a HiLoad
Superdex 75 column (GE Healthcare) equilibrated in 40 mM HEPES, pH 7.5,
100 mM NaCl. The purity of the protein was evaluated by gel electrophoresis and
the purified protein was stored at −80 °C.

The gene encoding luciferase, subcloned into a pET21b vector, was kindly
provided by Professor Campbell at the University of Texas at Dallas[43]. Clarified
lysate was prepared as described above for trigger factor and loaded onto a Ni-His
Trap column (GE Healthcare) equilibrated in lysis buffer. The column was washed
with lysis buffer and the His-tagged luciferase was eluted in lysis buffer with
imidazole. The eluate was concentrated and loaded onto a HiLoad Superdex 75
column (GE Healthcare) equilibrated in 40 mM HEPES, pH 7.5, 200 mM NaCl.
The purity of the protein was evaluated by gel electrophoresis and the purified
protein was stored at −80 °C. Purified rabbit GAPDH used in this study was
purchased from Sigma-Aldrich (G2267).

**Sedimentation velocity-analytical ultracentrifugation.** All sedimentation
velocity-analytical ultracentrifugation experiments were carried out in a Beckman
Optima XL-1 centrifuge, equipped with a four-hole An-60 Ti rotor (Beckman
Coulter) as previously described[53]. For GAPDH experiments, 57.5 μM of denatured
GAPDH was prepared in buffer A (0.1 M potassium phosphate, pH 7.5, 0.1 M KCl
and 1 mM tris(2-carboxyethyl)phosphine hydrochloride) with 3 M GdnHCl was
diluted 11.5-fold into the pre-chilled buffer A with or without trigger factor. The
final concentration of GAPDH was 5 μM. For the luciferase experiments, 100 μM
of denatured luciferase prepared in buffer A with 5 M urea was diluted 50-fold into
the pre-chilled buffer A with or without trigger factor. In all experiments, 420-μl
samples were loaded into two-sector epon centerpieces with a 1.2 cm path length.
Sedimentation velocity data of GAPDH and luciferase were obtained at 4 °C using
the absorbance optical system at 280 nm at 40,000 and 35,000 r.p.m., respectively.
Data analysis was conducted using UltraScan-III 4.0[54,55].

**Size-exclusion chromatography.** Size-exclusion chromatography experiments
were performed using a Superdex® 200 Increase 10/300 GL column (GE Health-
care). The sample preparation method was identical to the one described for
Sedimentation velocity-analytical ultracentrifugation experiments. 500-μl samples
were loaded onto the column equilibrated with buffer A containing 0.26 M
GdnHCl. The apparent molecular weight was estimated based on the standard
curve of gel filtration calibration kits (GE Healthcare).

**Protein labelling**. Fluorescein-5-maleimide-labelled trigger factor was obtained by incubating the purified trigger factor E326C with a 10-fold excess of fluorescein-5-maleimide (AAT Bioquest) in buffer A at room temperature for 2 h. The excess dye was then removed from the labelled proteins by using a PD-10 column (GE Healthcare). The labelling efficiency was estimated by comparing the dye concentration with the corrected protein concentration which was determined as the following equation, Eq. (1):

$$\text{Dye concentration} = \frac{A_{495}}{\varepsilon_{495}} * \text{dilution factor} \qquad (1)$$

Corrected protein concentration $= \frac{A_{280} - (A_{495} * \text{CF})}{\varepsilon_{280}} *$ dilution factor, where $\varepsilon_{495}$ is the extinction coefficient for fluorescein-5-maleimide, which is $80{,}000\ M^{-1}\ cm^{-1}$; $\varepsilon_{280}$ is the extinction coefficient for trigger factor, which is $17{,}420\ M^{-1}\ cm^{-1}$ and CF is the correction factor, which is 0.275 according to the product information. The labelling efficiency of fluorescein-5-maleimide-labelled trigger factor was typically above 92%.

The Alexa Fluor 488-labelled GAPDH was obtained by incubating the GAPDH protein with a 10-fold excess of CF$^{TM}$ Fluor 488 A maleimide (Sigma-Aldrich) in buffer A at room temperature for 2 h. Although rabbit GAPDH has four cysteine residues on its sequence, only one cysteine (Cys 149) is solvent-exposed in the native tetrameric state and can be carboxymethylated by reacting with iodoacetate[56]. The excess dye was then removed from the labelled proteins by using PD−10 column (GE Healthcare). The labelling efficiency was estimated by comparing the dye concentration with the corrected protein concentration which was determined as the following equation, Eq. (2):

$$\text{Dye concentration} = \frac{A_{490}}{\varepsilon_{490}} * \text{dilution factor} \qquad (2)$$

Corrected protein concentration $= \frac{A_{280} - (A_{490} * \text{CF})}{\varepsilon_{280}} *$ dilution factor, where $\varepsilon_{490}$ is the extinction coefficient for Alexa Fluor 488, which is $70{,}000\ M^{-1}\ cm^{-1}$; $\varepsilon_{280}$ is the extinction coefficient for GAPDH, which is $35{,}780\ M^{-1}\ cm^{-1}$ and CF is the correction factor, which is 0.1 according to the product information. The labelling efficiency of Alexa Fluor 488-labelled GAPDH was typically above 98%.

**Anisotropy titration experiments**. Anisotropy titration experiments were conducted on a Cary Eclipse Fluorescence Spectrophotometer. Binding of client protein was monitored by fluorescence anisotropy of fluorescein-5-maleimide-labelled trigger factor with an excitation wavelength of 496 nm and emission wavelength of 520 nm. For binding to refolding GAPDH, 0.5 μM labelled trigger factor in 1 ml of buffer A was titrated with GdnHCl-denatured GAPDH and fluorescence anisotropy was recorded at 4 °C; For binding to refolding luciferase, 0.5 μM labelled trigger factor in 1 ml of buffer A was titrated with urea-denatured luciferase and fluorescence anisotropy was recorded at 25 °C. For each titration, the anisotropy was recorded 20 times and averaged. The averaged anisotropy was plotted as a function of GAPDH or luciferase concentrations. To obtain the apparent dissociation rate constant ($K_d$), the GAPDH binding curve and the luciferase binding curve were fitted to a quadratic equation and to a hyperbola equation, respectively.

For monitoring the dimerization of trigger factor, 0.5 μM labelled trigger factor was prepared in 1 ml of buffer A, containing various concentrations of denaturants (GdnHCl or urea) and was titrated with non-labelled trigger factor. The dissociation rate constant for trigger factor dimerization was determined by fitting titration curves to a hyperbola equation.

**Stopped-flow kinetic experiments**. Kinetic experiments were performed on a KinTek SF-300X stopped-flow instrument (KinTek Corporation) at 4 °C. Kinetics of trigger factor binding to GAPDH were monitored via changes in fluorescence anisotropy. 1.5 μM denatured Alexa Fluor 488-labelled GAPDH was loaded into a 0.5-μl syringe and various concentrations of trigger factor were loaded into a 5-μl syringe. The final concentration after mixing labelled GAPDH was 0.13 μM and the final concentration of trigger factor was 0.13–3.24 μM. The fluorophore labelled GAPDH were excited at 496 nm, and emission was detected using a 520 ± 10 band-pass filter. The instrument correction factor ($G$ factor) was determined by rotating the excitation beam using a polarization rotator and recording the ratio between vertically and horizontally polarized emission intensities of labelled GAPDH. The anisotropy was then calculated using the following equation, Eq. (3):

$$r = \frac{I_{vv} - G I_{vh}}{I_{vv} + 2 G I_{vh}} \qquad (3)$$

where $r$ is anisotropy, $G$ is the instrument correction factor, $I_{vv}$ is the fluorescence intensity when using a vertical polarizer on the excitation and vertical polarizer on the emission, and $I_{vh}$ is the fluorescence intensity when using a vertical polarizer on the excitation and horizontal polarizer on the emission.

Kinetics of refolding of GAPDH in the presence of trigger factor was monitored via changes in intrinsic tryptophan fluorescence. Fluorescence of tryptophan of GAPDH was excited at 296 nm and the emission was collected using a 340 ± 10 band-pass filter. 16 μM denatured GAPDH was loaded into a 0.5-μl syringe and various concentrations of trigger factor W151F were loaded into a 5-μl syringe. The final concentration of GAPDH was 1.4 μM and the concentration of trigger factor W151F after mixing was 0–11.2 μM. 10 traces were averaged in the binding

experiments, and 4 traces were averaged in the folding experiment. All the averaged traces were fitted to a single-exponential curve using KaleidaGraph (Synergy Software). All the buffer conditions after mixing was 0.1 M potassium phosphate, pH 7.6, 0.1 M potassium chloride, 1 mM tris(2-carboxyethyl)phosphine hydrochloride and 0.26 M GdnHCl.

**In vitro GAPDH activity assay**. All GAPDH activity assays were performed in 96-well clear flat bottom polystyrene microplate (Corning, NY, USA). Reactivation of GAPDH was initiated by 11.5-fold diluting 32 μM denatured GAPDH into buffer A containing different concentrations of trigger factor. The final concentration of refolding GAPDH is 2.78 μM. The protein mixtures were incubated on ice, and the enzymatic activity was measured every 30 min by 100-fold diluting the protein mixture into the activity assay buffer (0.1 M potassium phosphate, pH 7.6, 0.1 M potassium chloride, 1 mM EDTA, 1 mM tris(2-carboxyethyl)phosphine hydrochloride, 1.5 mM b-NAD (Sigma-Aldrich) and 1 mM DL-glyceraldehyde-3-phosphate (Sigma-Aldrich)). The GAPDH activity reaction was monitored by the increase in the absorbance of NADH at 340 nm using a BMG FLUOstar Omega Microplate Reader (Ortenberg, Germany). The slope of the steepest linear part of the absorbance curve is proportional to the amount of functional GAPDH. The refolding yield was calculated by comparing the slope of the curve between refolding GAPDH and the same amount of native GAPDH (without denaturation).

To test the effects of DnaKJE system on GAPDH refolding, the DnaKJE system (20 μM DnaK, 4 μM DnaJ, 20 μM GrpE). 2.78 μM refolding GAPDH was first incubated with 11.12 μM trigger factor on ice for 3 h, and the refolding mixture was then 1:1 mixed with the DnaKJE system with and without 5 mM ATP and MgSO$_4$. The refolding solutions were continued to incubate at room temperature for another hour before measuring the enzymatic activity.

**Tryptophan fluorescence spectroscopy**. All fluorescence spectra were measured on a Cary Eclipse Fluorescence Spectrophotometer at 4 °C with an excitation wavelength of 296 nm and emission wavelengths scanning from 300 to 400 nm. The spectra of native GAPDH and GdnHCl-denatured GAPDH were recorded in buffer A containing 0.26 M GdnHCl and 3 M GdnHCl, respectively. The refolding GAPDH was prepared by 11.5-fold dilution of GdnHCl-denatured GAPDH into buffer A in the absence and in presence of trigger factor W151F. All the proteins were used at a final concentration of 3 μM. All the samples were incubated on ice for an hour before the measurements. Each spectrum was shown as an average of 10 scans. For kinetic experiments, the samples were prepared as described above and the tryptophans of GAPDH were excited at 296 nm and the emitted fluorescence signals were collected at 320 ± 5 nm. In all the experiments with trigger factor W151F, the background signal of trigger factor W151F was removed by subtracting the background signal of trigger factor W151F from the kinetic trace of GAPDH refolding in the presence of the same amount of trigger factor W151F.

**Circular dichroism spectroscopy**. Far-UV circular dichroism spectra were acquired on a Jasco J-1500 CD spectrometer at 4 °C using a quartz cell with a path length of 0.1 cm. The spectra of native and GdnHCl-denatured GAPDH were recorded in buffer A containing 0.06 M GdnHCl and 3 M GdnHCl, respectively. The refolding GAPDH was prepared by 50-fold diluting the GdnHCl-denatured GAPDH into buffer A in the absence or in the presence of the same amount of trigger factor. All the samples were incubated on ice for an hour before the measurements. In the experiments with trigger factor, the background signal of trigger factor was removed by subtracting the spectrum of trigger factor alone from the spectrum of refolding GAPDH in the presence of trigger factor. The final concentration of GAPDH in all cases was 8.4 μM. Each spectrum was shown as an average of 5 scans.

**Limited proteolysis with trypsin**. Refolding GAPDH was prepared by a 50-fold dilution of GdnHCl-denatured GAPDH into buffer A in the absence or in the presence of trigger factor. The final concentration of GAPDH was 5 μM. The refolding solutions were then either directly digested with trypsin or incubated on ice for an hour before digestion. Proteolysis reaction was conducted at room temperature with 12.5 μg/ml (after mixing) trypsin (Sigma-Aldrich) for a minute, then the reaction was terminated with 0.5 mM trypsin inhibitor (Sigma-Aldrich). The digested samples were analysed by 4–15% Mini-PROTEAN TGX stain-free protein gel.

**Chemical cross-linking mass spectrometry (XL-MS)**. 6 μM denatured GAPDH was diluted into buffer A containing an equimolar amount of trigger factor. The final GdnHCl concentration was 0.26 M GdnHCl. The mixture of trigger factor and GAPDH was incubated on ice for 2 h. 2 mM disuccinimidyl dibutyric urea (Thermo Fisher Scientific) was added to the protein mixture and the crosslinking reaction was left to proceed on ice for 2 h. The reaction was quenched by adding 20 mM Tris (pH 7.5). 5 micro-litres of crosslinked samples were analysed by SDS–PAGE to determine the efficiency of crosslinking. The remaining samples were precipitated using chloroform–methanol, as previously described[57]. Briefly, methanol (150 μl) and chloroform (50 μl) were added to a sample of the cross-linked proteins (50 μl, 6 μM). After vortexing to mix (30 s), water (100 μl) was

added. The solution was mixed again (by vortexing for 30 s) before being centrifuged ($10,000 \times g$, 2 min). The upper layer of aqueous solvent was removed, and methanol (150 µl) was added. The solution was mixed (by vortexing for 30 s) and centrifuged ($10,000 \times g$, 2 min), after which the supernatant was removed, and the precipitate was air-dried by placing it in a laminar flow hood. The protein pellet was resuspended in 1% (w/v) Rapigest (Waters), 50 mM ammonium bicarbonate pH 8 (10 µl). Dithiothreitol (50 mM in 50 mM ammonium bicarbonate pH 8, 10 µl) was then added and the mixture was incubated at 37 °C for 1 h. Iodoacetamide (100 mM in 50 mM ammonium bicarbonate pH 8, 10 µl) was then added and the mixture was incubated at 37 °C for a further 1 h. Subsequently, trypsin was added (1:50 w/w enzyme:protein, Promega) along with 50 mM ammonium bicarbonate pH 8 (70 µl) and the mixture was incubated overnight at 37 °C. The resultant peptides were desalted using Sep-Pak tC18 cartridges (Waters), according to the manufacturer's instructions. The desalted peptides were evaporated to dryness and resuspended in 0.1% trifluoroethanol (50 µl). The peptides were then analysed by liquid chromatography–mass spectrometry (LC–MS) on an Orbitrap Exploris 240 mass spectrometer (Thermo Fisher).

Peptides (3 µl) were injected onto a 30 cm capillary emitter column (inner diameter 75 µm, packed with 3-µm Reprosil-Pur 120 C18 media, Dr. Maisch) prepared in-house, and separated by gradient elution of 2–30 % (v/v) solvent B (0.1% (v/v) formic acid in acetonitrile) in solvent A (0.1% (v/v) formic acid in water) over 60 min at 300 nL min$^{-1}$. The mass spectrometer was operated in data-dependent acquisition mode with precursor fragmentation performed by Higher-energy C-trap dissociation. Each high-resolution scan (m/z range 500–1500, $R = 60,000$) was followed by 10 product ion scans ($R = 15,000$), using stepped normalised collision energies of 27%, 30% and 33%. Only precursor ions with charge states 3–7+ (inclusive) were selected for tandem MS. The dynamic exclusion was set to 60 s. Cross-link identification was performed using MeroX v2.0.1.4, considering Lys-Lys/Ser/Thr/Tyr cross-links. A maximum of 2 of the four marker ions, corresponding to fragmentation of the cross-linker, were allowed to be missing in the database search, the mass tolerances were set to 10 ppm and the false discovery rate was set to 1%. Manual verification of all spectra was performed to ensure correct assignment and that a significant degree of sequence coverage of each cross-linked peptide was present in the spectrum. Raw XL-MS data have been deposited to the ProteomeXchange Consortium the PRIDE partner repository with the dataset identifier PXD029365 A reporting summary (based on community guidelines[58]) can be found (Supplementary Data 1).

**Hydrogen–deuterium exchange mass spectrometry (HDX-MS).** For HDX-MS experiments, a robot for automated HDX (LEAP Technologies) was coupled to a Acquity M-Class LC and HDX manager (Waters). Samples comprised protein (TF, GAPDH or both, at a concentration of 8 µM) in 10 mM potassium phosphate, pH 7.6, 0.1 M potassium chloride, 1 mM tris(2-carboxyethyl)phosphine hydrochloride and 0.26 M GdnHCl. To initiate the HDX experiment, 95 µl of deuterated buffer (10 mM potassium phosphate, pD 7.6, 0.1 M potassium chloride, 1 mM tris(2-carboxyethyl)phosphine hydrochloride and 0.26 M GdnHCl) was added to 5 µl of protein-containing solution, and the mixture was incubated at 4 °C for 0.5, 2, 5, 10 or 30 min. For each time point and condition three replicate measurements were performed. The HDX reaction was quenched by adding 100 µl of quench buffer (10 mM potassium phosphate, 0.05% DDM, pH 2.2) to 50 µl of the labelling reaction.

The quenched sample (50 µl) was proteolyzed by flowing through immobilised pepsin and aspergillopepsin columns (Affipro) connected in series (20 °C). The produced peptides were trapped on a VanGuard Pre-column [Acquity UPLC BEH C18 (1.7 µm, 2.1 mm × 5 mm, Waters)] for 3 min. The peptides were separated using a C18 column (75 µm × 150 mm, Waters, UK) by gradient elution of 0–40% (v/v) acetonitrile (0.1% v/v formic acid) in H$_2$O (0.3% v/v formic acid) over 7 min at 40 µl min$^{-1}$.

Peptides were detected using a Synapt G2Si mass spectrometer (Waters) operating in HDMS$^E$ mode, with dynamic range extension enabled. IM separation was used to separate peptides prior to CID fragmentation in the transfer cell. CID data were used for peptide identification, and uptake quantification was performed at the peptide level. Data were analysed using PLGS (v3.0.2) and DynamX (v3.0.0) software (Waters). Search parameters in PLGS were as follows: peptide and fragment tolerances = automatic, min fragment ion matches = 1, digest reagent = non-specific, false discovery rate = 4. Restrictions for peptides in DynamX were as follows: minimum intensity = 1000, minimum products per amino acid = 0.3, max sequence length = 25, max ppm error = 5, file threshold = 3. The software Deuteros was used to identify peptides with statistically significant increases/decreases in deuterium uptake and to prepare Wood's plots[59]. The raw HDX-MS data have been deposited to the ProteomeXchange Consortium via the PRIDE partner repository with the dataset identifier PXD029364. A summary of the HDX-MS data, as recommended by reported guidelines[60] is shown in Supplementary Table 2.

**In vitro bacterial luciferase activity assay**. Purified luciferase was denatured in buffer A containing 5 M urea at room temperature for 2 h. The urea-denatured luciferase was then diluted 50-fold into buffer A containing various concentrations of trigger factor (0–10.6 µM). The final concentration of luciferase was 1.06 µM. The refolding solutions were incubated at room temperature, and aliquots (4 µl)

were withdrawn at various time points and diluted 25-fold into buffer A containing 50 µM FMN and 1 mM EDTA. 10 mM sodium dithionite was added to reduce FMN to FMNH2, and the solution was then 1:1 mixed with buffer A containing 0.1% sonicated decanol before the measurement. The bacterial activity assay was monitored by luminescence using a BMG FLUOstar Omega Microplate Reader (Ortenberg, Germany).

**Reporting summary**. Further information on research design is available in the Nature Research Reporting Summary linked to this article.

## Data availability

The source data underlying Figs. 1c, d, 2a, d, 3a, c, 6c, d and Supplementary Figs. 3–8 are provided as a Source Data file. Raw analytical ultracentrifugation data have been included as Supplementary dataset. The XL-MS and HDX-MS data generated in this study have been deposited to the ProteomeXchange Consortium via the PRIDE partner repository under the accession codes PXD029365 (XL-MS) and PXD029364 (HDX-MS). The crystal structures reported in the Supplementary Figs. 2, 9 and 10 are available in the PDB database under accession code 3GTY, 1J0X, 1W26. Source data are provided with this paper.

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

## Acknowledgements
We thank Rishav Mitra and Mark Dulchavsky for insightful suggestions and their critical reading of the manuscript. JCAB and KW are funded by the Howard Hughes Medical Institute. ANC and TCM acknowledge support of a Sir Henry Dale Fellowship (awarded to ANC) jointly funded by the Wellcome Trust and the Royal Society (Grant Number 220628/Z/20/Z), and ANC acknowledges the support of a University Academic Fellowship from the University of Leeds. SER holds a Royal Society Professorial Fellowship (RSRP/R1/211057). Funding from NIHR (NIHR200633), BBSRC (BB/M012573/1) and the Wellcome Trust (208385/Z/17/Z) enabled the purchase of mass spectrometry equipment.

## Author contributions
K.W. and J.C.A.B. conceived the original idea; S.E.R., A.N.C., and J.C.A.B. supervised the research; K.W. performed all the experiments in this study, except the MS experiments; T.C.M. and A.N.C. conducted the XL-MS and HDX-MS experiments and analysed the MS data. K.W., A.N.C., and J.C.A.B. wrote the manuscript.

## Competing interests
The authors declare no competing interests.

## Additional information

**Peer review information** *Nature Communications* thanks Argyris Politis, Pierre Goloubinoff and other anonymous Reviewer(s) to the peer review of this work. Peer reviewer reports are available.

