## [Peer Review File · Nature Communications]

REVIEWER COMMENTS

Reviewer #1 (Remarks to the Author):

The authors used a blend of biochemical and biophysical techniques to thoroughly investigate the role of TF in the refolding of GAPDH. Their results largely reiterate the already established role of TF in refolding of GAPDH (Huang, G.C., et al., Assisted folding of d-glyceraldehyde-3-phosphate dehydrogenase by trigger factor. *Protein Science*, 2000) (Haldar, Shubhasis et al., Trigger factor chaperone acts as a mechanical foldase. *Nature communications*, 2017). However, the authors went beyond the current state-of-the-art and with the help of two complementary techniques, XL-MS and HDX-MS, were able to identify precisely the sites and conformational dynamics of TF and GAPDH interaction. This study further helps to understand how TF or other ATP-independent chaperones in general, are able to interact with partially-folded clients, preventing their aggregation and misfolding.

Despite the clear elaboration of the results, some points still need further clarification, prior to publication:

- 1) All the experiments are done in denaturing conditions (0.26M GdnHCl), where TF mainly exists as a monomer. To what extent it is comparable to the native condition of TF?
- 2) In Fig 2c, in the presence of TF only 25% of the original activity of GAPDH is recovered. In the introduction the authors claim that, once released from TF the intermediate state of GAPDH can self-assemble into an active tetrameric state without requiring assistance from other chaperones. If this is the case then 25% regain in activity is surprisingly low.
- 3) In Fig 3, the TF mutant W151F is used for fluorescence experiments. However, there is no control to establish that it behaves in a similar manner as WT for GAPDH specifically. Refolding activity of GAPDH in the presence of this mutant could be a good control.
- 4) In Fig 4a, the band of cross-linked TF:GAPDH should logically intensify with increasing concentrations of cross linker as their individual bands are getting lighter but it is not the case. What is the reason behind it?

Overall, the manuscript is clearly written and well presented. It is likely to add to the existing knowledge in the field.

Reviewer #2 (Remarks to the Author):

Trigger factor (TG) is a prokaryote-specific chaperone that does not use ATP-hydrolysis to promote the early folding steps of nascent polypeptides, during their synthesis and following their exit from the ribosome. In vitro, TG has been shown, mainly by Deuerling et al., to act both as a “holdase” chaperone and as a “foldase” chaperone, by way of both preventing the aggregation of artificially pre-unfolded proteins and by favoring the native folding of small domains in proteins, as they sequentially emerge from the ribosome. This mechanism, which has been compared to that of an ice-cream scooper, has been suggested to explain how upon release from trigger factor, the near-natively folded domains of a client protein may rapidly self-associate into native oligomers and acquire an enzymatic activity, without needing additional folding factors.

Here, using Guanidium HCl denatured glyceraldehyde 3-phosphate dehydrogenase (GAPDH), Wu et al., brought additional in vitro evidence that trigger factor plays an active role in promoting the native folding of an artificially pre-unfolded client protein, by preventing non-productive self-associations (misfolding) and by shielding oligomeric interfaces from aggregation. The authors used enzyme kinetics and various biophysical methods, such as analytical ultracentrifugation, deuterium exchange, XL-MS and HDX-MS to identify the specific surfaces on TG that interact with specific surfaces on GAPDH. They showed that the state of the TG-bound client GAPDH was that of a near native monomer and that TG selectively shielded oligomeric interfaces of GAPDH. The authors main claim was that upon release from TG, the mostly natively folded monomeric GAPDH can rapidly self-associate into a native tetramer with enzymatic activity, without needing additional folding factors.

Wu et al., suggest a mechanism combining passive prevention of aggregation and more active catalytic folding. Yet, they did not address the details of this proposed catalytic mechanism, such as the precise structural changes taking place in the “holdase” and “foldase” catalytic sites of TG, during the various steps of catalysis, and in the client polypeptides while being converted from high-affinity inactive TG substrates, into high-affinity TG-bound near native intermediates, and finally turning by an unclear mechanism into low-affinity near-native products that spontaneously assemble into active tetramers.

Major issues:

1) The central claim that TG can catalyze folding is not new. A similar foldase activity has been initially shown by Mashaghi et al., 2013, who investigated individual maltose binding protein molecules using optical tweezers. They showed that TG binds folded structures smaller than one domain, which are then stable for seconds and ultimately convert to the native state. TG was shown to stimulate native folding in constructs of repeated MBP domains, indicating that it can promote the correct folding of substrates by protecting partially folded states from distant interactions that produce stable misfolded states.

2) This study is based on a single client protein: Guanidium-HCl-denatured GAPDH. One would have expected that for comparison, the same detection methods would be applied on other artificially denatured proteins, to highlight similarities, as well as differences of the interaction sites, of the deuterium exchange patterns, etc. What about the commonly used firefly luciferase? The absence of data on the effects, or lack of effects of TIG on the refolding of pre-denatured luciferase, raises the possibility that it may not behave similarly to GAPDH. Yet, it would have been very important to have luciferase, among others, as control to compare and valorize the data presented here solely for GAPDH.

Further, given that GAPDH is a mammalian enzyme and TG a prokaryote-specific chaperone, it would have been of considerable importance that this study included and used a natural *E. coli* client of TG. An unbiased search for *E. coli* TG clients could have been attempted by performing, for example, LIP-MS (see Cappelletti 2021 <https://doi.org/10.1016/j.cell.2020.12.021>) on wild type versus delta TG *E. coli* mutants, and/or on delta TG *E. coli* with down-regulated expression of DnaKJE.

3) This brings another unaddressed question: the collaboration between TG and DnaKJE. It has been shown more than two decades ago, by the groups of Bukau and Hartl, that in the *E. coli* cell, TG closely collaborates with the ATP-dependent DnaK chaperone system, in promoting the proper folding of nascent proteins. Whereas knockout mutants either of DnaK/DnaJ or of TG could survive and grow at low temperatures, the double mutant was synthetic lethal. Thus, the functionality of TG as a co-chaperone of DnaK and vis-versa of DnaK as a co-chaperone of TG (as it is likely the case with SSB and SSZ in the cytosol of eukaryotes), is central to the understanding of the mechanism by which TG acts in prokaryotes, both as an anti-aggregation and as a pro-foldase enzyme.

4) TG is sequence-inferred to be also a PPIase. This is a central issue when it comes to address the pro-foldase function of TG. But the potential PPIase function of TG has not been addressed here. There have been in the literature of PPIases, protein substrates whose refolding to the native state is limited by the slow rate of spontaneously cis-trans isomerization of their prolines, a slow rate that can be strongly accelerated by PPIases. The authors would have been well-advised to use a model reporter enzyme, whose rate of native refolding would have been chosen to particularly depend on proline cis-trans isomerization. Note that many model chaperone client enzymes, when pre-incubated with 8M urea or 6M Guanidinium-HCl show, following dilution, different yields of subsequent refolding, without and with chaperones: longer pre-incubation often reduce the rates and yields of refolding, suggesting that the complete pre-isomerization of all the prolines in the presence of Guanidinium-HCl take minutes, and not milliseconds. How long were the GAPDH pre-incubated before dilution for the TG assays ?

By the way, Guanidinium HCl is a salt. Compared to the ~150 mM of K⁺ and ~10 mM Mg²⁺ with Cl⁻ and various organic anions in the *E. coli* cell, to which evolution has optimally accustomed TG, 260 mM of residual Guanidinium HCl is not a negligible amount of an unnatural salt. Similar amounts of salt have been shown to inactivate the ClpB disaggregase, and Na⁺ has been shown to inhibit DnaK.

Urea in contrast, is not a salt. It is more natural and can be found in dozens of mM in cells of sharks. Thus, to exclude that the 260 mM Guanidinium HCl affected the in vitro observations reported here, the same experiments should be repeated with GAPDH pre-denatured with 8 M urea.

Minor issues:

1) In their introduction, the authors state that there are only two types of chaperones ATP-independent "holdases" and ATP-dependent "foldases". Yet, in the last two decades, there was growing evidence that some chaperones, such as GroEL and Hsp70 and Hsp104, are in fact catalytic unfoldases, which by way of decompacting stable misfolded species, consequently accelerate the native spontaneous refolding of proteins. Although strongly opposed to this view for decades, FU Hartl recently endorsed an unfolding mechanism for HSP70 (Rahmi Imamoglu 2020 doi: 10.1038/s41467-019-14245-4). It is surprising that the authors did not discuss in their introduction and discussion, the possibility that some

foldases, including TG, could be catalytic unfoldases. Did the authors consider addressing this question experimentally, for example by checking if upon initial binding to TG, the pre-unfolded and/or already misfolded GAPDH species do not transiently become more sensitive to limited amounts of trypsin or proteinase-K, before reaching later a near native state expected to be more resistant to proteases ?

2) The authors use the term “holdase” nine times in their manuscript. It is intriguing that experts of the molecular chaperone field use this term to describe a molecular function that is not catalytic. In contrast, enzymologists have long agreed to add the suffix “ase” only to bona fide enzymes, i.e. to protein catalysts that can accelerate the conversion of a large excess of high-affinity substrates into low-affinity products and that enzymes should not be part of the final product of their catalysis, as John Ellis so rightfully emphasized in his seminal definition of the molecular chaperones.

Reviewer #3 (Remarks to the Author):

Trigger factor is known to be a ribosome associated folding factor. In the present study, authors present results for an active foldase function of trigger factor for unfolded GAPDH which requires proper assembly into a tetramer state to be active. The authors use an impressive set of advanced biophysical techniques to gain novel insight into the chaperone function of trigger factor. They show that trigger factor binds to a partially folded state of GAPDH upon its dilution from the denaturant. Folding seems to occur to some extent in the trigger factor bound state and GAPDH becomes enzymatically active after release from the complex.

This is an interesting concept. However there are some open questions concerning aspects of the study.

One question concerns the affinity of trigger factor for GAPDH. From thermophoresis authors conclude it to be 45 nM. This is a high affinity interaction and one wonders how this allows folding of GAPDH in complex with trigger factor. However, the stopped flow anisotropy experiments suggest rather a micromolar KD. This needs to be resolved.

In the refolding assays, 0.26 M GdnHCl is used. At this concentration trigger factor is a monomer as shown by authors. However, they state that physiologically trigger factor exists as a dimer. Authors may want to reduce the GdnHCl concentration to reach conditions where trigger factor is a dimer. Does this influence the results?

Are differences observed with time in the HDX pattern of GAPDH when in complex with trigger factor? It will be interesting to see if GAPDH undergoes structural fluctuations while bound to trigger factor. The overall structure of GAPDH is shown to be dynamic in complex with the trigger factor. It is stated that trigger factor primarily binds to the oligomerization interface and aggregation prone regions of GAPDH. These regions are shown to undergo maximum deuterium uptake in HDX and are most dynamic in

complex with trigger factor. Shouldn't they be more protected from exchange if they are tightly bound to trigger factor?

How is GAPDH released from trigger factor? For the enzymatic activity, very high dilutions may trigger release of GAPDH. Does aggregation occur under these conditions upon spontaneous refolding? Are such high dilutions physiologically relevant or are these studies meant to demonstrate basic properties of the complex?

The characterization of GAPDH bound to trigger factor by fluorescence provides interesting insight into the folding state. As this is one of the core findings of the paper additional results to support and extent the conclusion that it is a near native state would further strengthen the study.

For these experiments a trigger factor variant lacking the intrinsic Trp residue is used. I am sure the authors have performed experiments shown for wt trigger factor also with this variant. These should be included in the supplement.

An effort should be made to integrate the results of previous studies on trigger factor and GAPDH (Huang et al.; Liu et al.) in more detail into the discussion of the current results.

Minor points

Error bars are missing from many graphs.

What is the y-axis shown in Fig. 1c?

In Fig. 1d, extrapolation to zero time point should be shown in a zoomed view. The corresponding figure in the text is mentioned as Fig. 1e when it should be Fig. 1d.

Fig. 2c should show a control of native GAPDH activity at similar concentrations used for refolded GAPDH. Why does the increase in absorbance in Fig. 2c not saturate?

In Fig. 2d, a control of GAPDH alone and a native mix of GAPDH and trigger factor should be shown.

The fluorescence values on y-axis for Fig. 3a do not seem to be absolute values as shown in Fig. 3d.

In the legend to Fig. 5, authors should mention the time point for which the HDX data is shown.

We have now addressed all the reviewer's comments in an extensively revised manuscript that incorporates significant new data, including the analysis of the effect of trigger factor on the refolding and binding of a client protein which trigger factor assists the folding of *in vivo* substrate. The main changes to the manuscript are highlighted in green, the introduction is 65% new, the results and discussion are 46% new and the conclusions are 39% new. We hope that the manuscript is now acceptable for publication.

Reviewer #1 (Remarks to the Author):

The authors used a blend of biochemical and biophysical techniques to thoroughly investigate the role of TF in the refolding of GAPDH. Their results largely reiterate the already established role of TF in refolding of GAPDH (Huang, G.C., et al., Assisted folding of d-glyceraldehyde-3-phosphate dehydrogenase by trigger factor. Protein Science, 2000) (Halder, Shubhasis et al., Trigger factor chaperone acts as a mechanical foldase. Nature communications, 2017). However, the authors went beyond the current state-of-the-art and with the help of two complementary techniques, XL-MS and HDX-MS, were able to identify precisely the sites and conformational dynamics of TF and GAPDH interaction. This study further helps to understand how TF or other ATP-independent chaperones in general, are able to interact with partially folded clients, preventing their aggregation and misfolding.

We thank the reviewer for their positive comments!

Despite the clear elaboration of the results, some points still need further clarification, prior to publication:

As listed below we have now addressed all the points that the reviewer considered to require further elaboration.

1) All the experiments are done in denaturing conditions (0.26M GdnHCl), where TF mainly exists as a monomer. To what extent it is comparable to the native condition of TF?

To address the reviewer's question of if denaturant induced monomerization affects the results obtained, we first measured the K_d for TF dimerization under various buffer conditions including those that do not contain any of the denaturant using anisotropy titration experiments. We have now showed that the K_d for TF dimerization under native conditions (0.1 M KPi, 0.1 M KCl and 1 mM TCEP) was 18.4 ± 0.8 μ M. Addition of various concentrations of denaturant differently effects TF dimerization. The values of K_d measured in different concentrations of denaturant are shown in the table below, which are now included the revised supplementary Fig. 3a.

[denaturant]	K_d (μ M)
Native buffer	18.4 ± 0.8
+0.06 M GdnHCl	29 ± 4
+0.26 M GdnHCl	76 ± 6
+0.16 M urea	35 ± 3
+0.7 M urea	85 ± 4

Importantly, while we see different degrees of denaturant effects on TF dimerization under various conditions, trigger factor's effect on GAPDH refolding was not substantially affected, suggesting that TF dimerization does not interfere with client binding (see the data in the supplementary Fig. 3b-e). This is somewhat surprising given that the dimeric interface of TF largely overlaps with its client binding surface (Saio, T., eLife, 2018; 7:e35731.). However, given that the K_d of trigger factor binding to refolding GAPDH (~ 0.3 μ M) is more than 50 fold lower than the K_d for TF dimerization (~ 18 μ M), we reason that the presence of client should readily trigger TF monomerization, allowing monomeric trigger factor to associate with its clients.

2) In Fig 2c, in the presence of TF only 25% of the original activity of GAPDH is recovered. In the introduction the authors claim that, once released from TF the intermediate state of GAPDH can self-assemble into an active tetrameric state without requiring assistance from other chaperones. If this is the case then 25% regain in activity is surprisingly low.

Trigger factor is known to cooperate with the KJE system and GroEL/ES to assist protein folding. Here, we reported an alternative pathway by which trigger factor assists protein folding/assembly in the absence of downstream ATP-dependent chaperones. Though we postulated that the intermediate state of GAPDH *can* self-assemble into an active tetrameric state without requiring assistance from other chaperones, given this intriguing comment from the reviewer we wondered if the refolding yield of GAPDH could be further increased by the addition of ATP-dependent chaperones. To test this, we incubated the refolding GAPDH with an excess of trigger factor on ice for 2 hours. Then, DnaKJE was added to the solution and the mixture was incubated at room

temperature for another 1 hour prior to the measurement of enzymatic activity. We found that the addition of DnaKJE could further increase the refolding yield from 35% to 50% (see new Supplementary Fig. 5). These results indicated that a fraction of GAPDH is not able to fold or assemble correctly along the trigger factor-mediated folding pathway. Some of these GAPDH molecules may form into a kinetically trapped intermediate state that requires the assistance of ATP-dependent chaperones to further promote their folding.

Note that the previously reported 25% yield appears to be an inaccurate value due to a calculation error on our part. After recalculating the refolding yield from the same set of the data, the actually refolding yield from this experiment is ~35%. We have clarified this on the page 6 of the revised manuscript and have included the trace for the activity of native GAPDH in Figure 2b for direct comparison.

3) In Fig 3, the TF mutant W151F is used for fluorescence experiments. However, there is no control to establish that it behaves in a similar manner as WT for GAPDH specifically. Refolding activity of GAPDH in the presence of this mutant could be a good control.

We have now compared the chaperone activities of WT trigger factor and the W151F variant in our GAPDH refolding assay and found that they perform a very similar manner. These new data are now available in a new Supplementary Figure 6.

4) In Fig 4a, the band of cross-linked TF:GAPDH should logically intensify with increasing concentrations of cross linker as their individual bands are getting lighter but it is not the case. What is the reason behind it?

In Figure 4a we show that even at the lowest concentration of added crosslinker, the monomeric GAPDH band almost completely disappears. The TF band also disappears as a function of the concentration of added crosslinker. Above the gel band corresponding to the crosslinked complex there is also a broad 'smear', corresponding to higher molecular weight species. The intensity of this increases with crosslinker concentration, suggesting that this may be the reason why the band corresponding to the crosslinked complex slightly decreases in intensity in the 2 mM DSBU sample. We have clarified this in the figure legend.

Overall, the manuscript is clearly written and well presented. It is likely to add to the existing knowledge in the field.

We thank the reviewer for these kind comments.

Reviewer #2 (Remarks to the Author):

Trigger factor (TG) is a prokaryote-specific chaperone that does not use ATP-hydrolysis to promote the early folding steps of nascent polypeptides, during their synthesis and following their exit from the ribosome. In vitro, TG has been shown, mainly by Deuerling et al., to act both as a “holdase” chaperone and as a “foldase” chaperone, by way of both preventing the aggregation of artificially pre-unfolded proteins and by favoring the native folding of small domains in proteins, as they sequentially emerge from the ribosome. This mechanism, which has been compared to that of an ice-cream scooper, has been suggested to explain how upon release from trigger factor, the near-natively folded domains of a client protein may rapidly self-associate into native oligomers and acquire an enzymatic activity, without needing additional folding factors.

Here, using Guanidium HCl denatured glyceraldehyde 3-phosphate dehydrogenase (GAPDH), Wu et al., brought additional in vitro evidence that trigger factor plays an active role in promoting the native folding of an artificially pre-unfolded client protein, by preventing non-productive self-associations (misfolding) and by shielding oligomeric interfaces from aggregation. The authors used enzyme kinetics and various biophysical methods, such as analytical ultracentrifugation, deuterium exchange, XL-MS and HDX-MS to identify the specific surfaces on TG that interact with specific surfaces on GAPDH. They showed that the state of the TG-bound client GAPDH was that of a near native monomer and that TG selectively shielded oligomeric interfaces of GAPDH. The authors main claim was that upon release from TG, the mostly natively folded monomeric GAPDH can rapidly self-associate into a native tetramer with enzymatic activity, without needing additional folding factors.

Wu et al., suggest a mechanism combining passive prevention of aggregation and more active catalytic folding. Yet, they did not address the details of this proposed catalytic mechanism, such as the precise structural changes taking place in the “holdase” and “foldase” catalytic sites of TG, during the various steps of catalysis, and in the client polypeptides while being converted from high-affinity inactive TG substrates, into high-affinity TG-bound near native intermediates, and finally turning by an unclear mechanism into low-affinity near-native products that spontaneously assemble into active tetramers.

A long-standing problem in the chaperone field is that it remains difficult to determine the “precise structural changes” taking place in both the chaperone and client proteins. We recognize the importance of this challenge. Indeed, we have been able to partially achieve these goals for the Spy-Im7 chaperone-client pair (Horowitz, S., NSMB, 2016; 23(7): 691–697.). This visualization however was dependent on the small size of both the chaperone (Spy), and client (Im7) and the unusual ease with which the complex could be crystallized. None of these apply to the trigger factor GAPDH complex, making it difficult to understand the precise structural changes going on during the TF chaperone mediated folding of GAPDH using the approaches we have developed. However, to obtain more structural information about this chaperone mediated folding process than were present in the original submission, we have added a number of new results, including Trp fluorescence, circular dichroism and limited proteolysis, all of which provide evidence for the conformational change of GAPDH

while bound to trigger factor (see new Figure 3). Together, these new data strengthen our argument that non-native GAPDH undergoes a conformational change while it is bound to trigger factor.

Major issues:

1) The central claim that TG can catalyze folding is not new. A similar foldase activity has been initially shown by Mashaghi et al., 2013, who investigated individual maltose binding protein molecules using optical tweezers. They showed that TG binds folded structures smaller than one domain, which are then stable for seconds and ultimately convert to the native state. TG was shown to stimulate native folding in constructs of repeated MBP domains, indicating that it can promote the correct folding of substrates by protecting partially folded states from distant interactions that produce stable misfolded states.

While the foldase activity of trigger factor has been shown by Mashaghi et al using optical tweezers, the structural and kinetic mechanism of how trigger factor “folds” its clients has previously remained largely unknown. Our current study provides new structural and kinetic information regarding how trigger factor associates with partially folded intermediates, and what role(s) trigger factor play during protein folding.

In the previous study cited by the reviewer, the authors demonstrated that a partially folded state of the client, which is less abundant during the spontaneous folding, is populated in the presence of trigger factor. The authors, however, were not able to show if the folding reaction occurs while bound to trigger factor or occurs upon release from the surface of trigger factor and how tightly trigger factor associates with its client protein while it facilitates the folding reaction. Here, we have obtained the missing kinetic and structural information about this process, by showing that trigger factor can allow a non-native protein to fold into a partially folded state while bound. Furthermore, we have used XL-MS and HDX-MS approaches to reveal the interface of the trigger factor-GAPDH complex and the structural dynamics of GAPDH bound to trigger factor. Our data suggest that trigger factor shields the aggregation-prone inter-domain interface to prevent non-productive interactions. Lastly, we propose a mechanism by which trigger factor assist refolding/reassembly of an oligomeric protein which hasn't been explored in previous studies using a monomeric protein, i.e. MBP.

2) This study is based on a single client protein: Guanidium-HCl-denatured GAPDH. One would have expected that for comparison, the same detection methods would be applied on other artificially denatured proteins, to highlight similarities, as well as differences of the interaction sites, of the deuterium exchange patterns, etc. What about the commonly used firefly luciferase? The absence of data on the effects, or lack of effects of TIG on the refolding of pre-denatured luciferase, raises the possibility that it may not behave similarly to GAPDH. Yet, it would have been very important to have luciferase, among others, as control to compare and valorize the data presented here solely for GAPDH.

To address this concern, we explored several other clients and we settled on making a concentrated effort to explore the effects of trigger factor on the reactivation of luciferase and explore its binding to this client. We decided to use *Vibrio harveyi* luciferase as a client instead of the suggested firefly luciferase for two reasons. First, as the reviewer notes, it has previously been shown that trigger factor has little effect on the folding of firefly luciferase, making it an unappealing target (Melkina, O.E., *Biochemistry (Moscow)*, 2014; 79 (1), 62-68). Second, we were excited to realize that trigger factor is known to affect the refolding processes of this bacterial luciferase both in vitro and in vivo (Melkina, O.E., *Biochemistry (Moscow)*, 2014; 79 (1), 62-68; Shieh, *Science*, 2015; 350(6261), 678-80). Thus, by using this client we could also address the reviewer's next suggestion that we find and explore the in vitro effect of trigger factor on the folding of an in vivo client.

In the absence of trigger factor, *V. harveyi* luciferase self-assembles into multiple non-productive oligomeric species during refolding, as analyzed by analytical ultracentrifugation (new Fig. 6a). This is similar to what we see in the spontaneous folding of GAPDH, (Fig.1a), and similarly opens up an avenue for exploring the effects of trigger factor on the refolding/assembly process.

Then, we showed that trigger factor binds to refolding luciferase with an apparent K_d of $\sim 4 \mu\text{M}$ (Fig. 6c). While this is weaker than the apparent K_d for binding GAPDH ($\sim 0.3 \mu\text{M}$), it is tighter than the apparent K_d for TF dimerization ($\sim 18 \mu\text{M}$). Next, we tested the effects of trigger factor on luciferase refolding. In the presence of trigger factor, a 1:1 complex between trigger factor and luciferase forms and the non-productive inter-subunit interactions within the luciferase population are largely diminished (Fig. 6b). Likely as a consequence, we found that trigger factor could further promote luciferase refolding/assembly and significantly increase its refolding yield as measured by an activity assay (Fig. 6d).

Further, given that GAPDH is a mammalian enzyme and TG a prokaryote-specific chaperone, it would have been of considerable importance that this study included and used a natural *E. coli* client of TG. An unbiased search for *E. coli* TG clients could have been attempted by performing, for example, LIP-MS (see Cappelletti 2021 <https://doi.org/10.1016/j.cell.2020.12.021>) on wild type versus delta TG *E. coli* mutants, and/or on delta TG *E. coli* with down-regulated expression of DnaKJE.

Fortunately, this type of unbiased search for in vivo trigger factor clients has already been done by others (Deuerling, E., 2003, Mol. Microbiol, 47(5), 1317-1328; Maisonneuve, E., J. Bacteriol., 2008, 190(3), 887-93; Martinez-Hackert, E., Cell, 2009, 138(5), 923-34. We highlight these observations in our revised manuscript. We note that although bacterial luciferase from *Vibrio harveyi* is an in vivo client for *E. coli* trigger factor as studied in *E. coli*, it might not be considered a “*natural E. coli* client for TG” but it certainly comes closer than the majority of clients used in chaperone studies historically. Although in an ideal world, scientists would only use natural clients for their chaperone studies, in the real world, the vast majority of studies use non-physiologic clients. The choice of chaperone-client pairs is largely determined by the limitations of the approaches selected. For instance, firefly luciferase, citrate synthase and malate dehydrogenase are commonly used in the “standard” aggregation assays; SH3 domains and peptides derived from alkaline phosphatase are often used in NMR studies; maltose-binding protein is often used in single molecule optical tweezers. These choices are driven in part because relatively few client proteins have the right combination of biophysical characteristics to enable their folding pathways to be well characterized, a necessary step if one wants to see how the chaperone affects client protein folding.

Here, we have tested the foldase effects of trigger factor on two very different client proteins, rabbit GAPDH and bacterial luciferase. In both cases, trigger factor can effectively prevent them from non-productive inter-subunit interactions and actively enhance their refolding yield.

3) This brings another unaddressed question: the collaboration between TG and DnaKJE. It has been shown more than two decades ago, by the groups of Bukau and Hartl, that in the *E. coli* cell, TG closely collaborates with the ATP-dependent DnaK chaperone system, in promoting the proper folding of nascent proteins. Whereas knockout mutants either of DnaKDnaJ or of TG could survive and grow at low temperatures, the double mutant was synthetic lethal. Thus, the functionality of TG as a co-chaperone of DnaK and vis-versa of DnaK as a co-chaperone of TG (as it is likely the case with SSB and SSZ in the cytosol of eukaryotes), is central to the understanding of the mechanism by which TG acts in prokaryotes, both as an anti-aggregation and as a pro-foldase enzyme.

We agree with the reviewer that it is interesting to understand how trigger factor cooperates with DnaKJE system. The present manuscript, however, focuses on the role TF plays independent of the KJE system and an extensive analysis of exactly how TF interacts with the KJE system is outside the scope of this manuscript. However, to address the question of if TF interacts with DnaKJE, we have performed new experiments showing that the addition of the DnaKJE can further promote TF-mediated GAPDH folding (Supplementary Fig. 5).

4) TG is sequence-inferred to be also a PPlase. This is a central issue when it comes to address the pro-foldase function of TG. But the potential PPlase function of TG has

not addressed here. There have been in the literature of PPlases, protein substrates whose refolding to the native state is limited by the slow rate of spontaneously cis-trans isomerization of their prolines, a slow rate that can be strongly accelerated by PPlases. The authors would have been well-advised to use a model reporter enzyme, whose rate of native refolding would have been chosen to particularly depend on proline cis-trans isomerization.

Indeed, trigger factor has PPlase activity that could in principle accelerate the folding of some slow-folding model proteins, like RCM-RNase T1. However, the PPlase activity at least so far seems to be of little practical importance as the TF F198A mutant, which has no PPlase activity, has little effect on its foldase activity in the *in vitro* GAPDH refolding and the *in vivo* chaperone activity (Kramer, G., JBC, 2004, 279(14), 14165-70). In addition, the deletion of PPlase domain has been shown to have little effect on chaperone activity both *in vitro* and *in vivo* (Merz, F., JBC, 2006, 281(42), 31963-71). In line with these results, we found that only one out of seven crosslinked sites is found on the PPlase domain. For these reasons we have not pursued the role of PPlase activity on GAPDH folding.

Note that many model chaperone client enzymes, when pre-incubated with 8M urea or 6M Guanidium-HCl show, following dilution, different yields of subsequent refolding, without and with chaperones: longer pre-incubation often reduce the rates and yields of refolding, suggesting that the complete pre-isomerization of all the prolines in the presence of Guanidium-HCl take minutes, and not milliseconds. How long were the GAPDH pre-incubated before dilution for the TG assays?

All samples of GAPDH were pre-incubated in denaturing buffer with 3M GdnHCl or 8M urea on ice overnight, which should enable their spontaneous isomerization toward the most energetically favorable form in the denatured condition (almost certainly the trans configuration). We note that there are no cis prolines present in the native crystal structures of GAPDH, suggesting that proline isomerization is not an issue in GAPDH refolding. Indeed, Huang et al., examined the foldase effect of trigger factor on long-term (overnight) denatured GAPDH and short-term (10s) denatured GAPDH and found no significant difference (Huang, Protein Science, 2000; 9, 1254-61).

By the way, Guanidium HCl is a salt. Compared to the ~150 mM of K⁺ and ~10 mM Mg²⁺ with Cl⁻ and various organic anions in the *E. coli* cell, to which evolution has optimally accustomed TG, 260 mM of residual Guanidium HCl is not a negligible amount of an unnatural salt. Similar amounts of salt have been shown to inactivate the ClpB disaggregase, and Na⁺ has been shown to inhibit DnaK.

Urea in contrast, is not a salt. It is more natural and can be found in dozens of mM in cells of sharks. Thus, to exclude that the 260 mM Guanidium HCl affected the *in vitro* observations reported here, the same experiments should be repeated with GAPDH pre-denatured with 8 M urea.

We have done as the reviewer suggested, namely to see if a residual amount of GdnHCl affected our *in vitro* observations. We have determined the ability of trigger

factor to refold GAPDH activity under various conditions, i.e., 0.26 M GdnHCl, 0.06 M GdnHCl, 0.7 M urea and 0.16M urea (new Supplementary Fig. 3b-e). Although the spontaneous refolding yield of GAPDH varies somewhat under these different conditions, trigger factor promotes the refolding of GAPDH in a very similar manner under all these conditions.

Minor issues:

1) In their introduction, the authors state that there are only two types of chaperones ATP-independent “holdases” and ATP-dependent “foldases”. Yet, in the last two decades, there was growing evidence that some chaperones, such as GroEL and Hsp70 and Hsp104, are in fact catalytic unfoldases, which by way of decompacting stable misfolded species, consequently accelerate the native spontaneous refolding of proteins. Although strongly opposed to this view for decades, FU Hartl recently endorsed an unfolding mechanism for HSP70 (Rahmi Imamoglu 2020 doi: 10.1038/s41467-019-14245-4). It is surprising that the authors did not discuss in their introduction and discussion, the possibility that some foldases, including TG, could be catalytic unfoldases. Did the authors consider addressing this question experimentally, for example by checking if upon initial binding to TG, the pre-unfolded and/or already misfolded GAPDH species do not transiently become more sensitive to limited amounts of trypsin or proteinase-K, before reaching later a near native state expected to be more resistant to proteases?

We thank the reviewer for this suggestion. We have added a section to discuss a possible catalytic unfoldase activity of trigger factor in our introduction. The chaperones with unfoldase activity that reviewer mentioned above are all ATP-dependent chaperones that use ATP binding or hydrolysis in some cases to guide protein folding and in other cases to unfold misfolded proteins, allowing them to reach their native states. Because these folding/unfolding processes usually involve kinetically-trapped states, foldase/unfoldase activities require energy, helping to explain their ATP-dependency. In contrast, ATP-independent chaperones are generally assumed not to be directly involved in protein folding or unfolding processes. Interestingly, Hoffmann, A., et al. (Hoffmann, A., *Cell*, 2012; 48(1):63-74), showed that trigger factor could bind and unfold pre-existing folded structure, though the unfoldase activity of trigger factor is limited by the thermodynamic stability of client protein, with trigger factor being apparently only able to unfold proteins with relatively low thermodynamic stability. We assumed from this study that it might be very challenging for trigger factor to unfold a misfolded protein, since these often exist in a stable kinetically-trapped state. However, inspired by the reviewer’s suggestion and pioneering work done by Pierre Goloubinoff and colleagues (Sharma, S.K., *Nat. Chem. Biol.*, 2010; 6(12):914-20), we decided to test if trigger factor is capable of unfoldase activity to assist in the refolding of its client proteins. To do this, urea-denatured luciferase was first diluted into refolding buffer and incubated at room temperature for an hour. During the incubation, refolding luciferase becomes protease-resistance, presumably by forming into a non-native state (Fig. Aa). Next, a 10-fold excess of trigger factor was added into refolding solutions and the activity of luciferase was measured at various time points. If trigger factor unfolds

misfolded luciferase prior to the refolding of luciferase, a delayed phase should be observed. Instead, we saw a slight increase in the refolding yield upon addition of trigger factor but no further increase in relative activity has been observed (Fig.Ab). In contrast, for a genuine unfoldase such as DnaK, it has been shown that DnaK can unfold a misfolded proteins prior to allowing them to refold to their native state (Sharma, S.K., Nat. Chem. Biol., 2010; 6(12):914-20). Our preliminary data suggest that trigger factor does not resolve pre-existing misfolded luciferase to facilitate protein refolding process. Instead, it appears that trigger factor, like other ATP-dependent chaperones, mainly functions in the stage prior to the misfolding or aggregation of client proteins.

Fig. A (a) Limited proteolysis analysis of refolding luciferase. 2 μM refolding luciferase was either directly digested with 12.5 $\mu\text{g}/\text{ml}$ trypsin from 1 min or incubated at room temperature for 1hr before trypsin digestion. Samples were analyzed by 4-15% Mini-PROTEAN TGX stain-free gels. (b) Bacterial luciferase activity was measured at various time points after the addition of 20 μM trigger factor into the 1 hr-aged refolding luciferase solution (red) or the addition of buffer (black).

2) The authors use the term “holdase” nine times in their manuscript. It is intriguing that experts of the molecular chaperone field use this term to describe a molecular function that is not catalytic. In contrast, enzymologists have long agreed to add the suffix “ase” only to bona fide enzymes, i.e. to protein catalysts that can accelerate the conversion of a large excess of high-affinity substrates into low-affinity products and that enzymes should not be part of the final product of their catalysis, as John Ellis so rightfully emphasized in his seminal definition of the molecular chaperones.

We agree and have replaced the term holdase with holding chaperone

Reviewer #3 (Remarks to the Author):

Trigger factor is known to be a ribosome associated folding factor. In the present study, authors present results for an active foldase function of trigger factor for unfolded GAPDH which requires proper assembly into a tetramer state to be active. The authors use an impressive set of advanced biophysical techniques to gain novel insight into the chaperone function of trigger factor. They show that trigger factor binds to a partially folded state of GAPDH upon its dilution from the denaturant. Folding seems to occur to some extent in the trigger factor bound state and GAPDH becomes enzymatically active after release from the complex.

We thank the reviewer for their kind comments!

However there are some open questions concerning aspects of the study.

One question concerns the affinity of trigger factor for GAPDH. From thermophoresis authors conclude it to be 45 nM. This is a high affinity interaction and one wonders how this allows folding of GAPDH in complex with trigger factor. However, the stopped flow anisotropy experiments suggest rather a micromolar KD. This needs to be resolved.

We admit that the Kd of 45 nM is surprisingly low. In the submitted manuscript this value was determined using microscale thermophoresis. One issue of using microscale thermophoresis was that the measurement was done at room temperature where GAPDH can aggregate. Because the reviewer's and our concerns, we decided to conduct anisotropy titration experiments, which is a more recognized approach than MST, and to conduct these experiments at 4C, a temperature at which no protein aggregation is observed – conditions that are more relevant to the functional studies we have performed. Using this approach, we were able to determine the apparent Kd to be 0.32 +/- 0.08 (uM).

To obtain the kinetic parameters we fitted the observed rate constant vs trigger factor concentration to a linear equation. In the previous submission we mistakenly included data points that were not collected under pseudo-first order conditions, which requires that the concentration of trigger factor be much larger than the concentration of GAPDH. If we replotted the curve using only data points with a 7-25 fold excess of trigger factor over GAPDH and fitted the plot to a linear equation we could determine more accurate values of kon and koff ($7.6 \cdot 10^6 \text{ M}^{-1} \text{ s}^{-1}$ and 6.7 s^{-1} , respectively) (the original kon and koff value are $6.6 \cdot 10^6 \text{ M}^{-1} \text{ s}^{-1}$ and 8.7 s^{-1} , respectively). Using these new values we can calculate a new Kd value of 0.88 uM which is fairly close to the Kd value that we obtained in the anisotropy titration experiment (with a Kd of 0.32 uM).

In the refolding assays, 0.26 M GdnHCl is used. At this concentration trigger factor is a monomer as shown by authors. However, they state that physiologically trigger factor exists as a dimer. Authors may want to reduce the GdnHCl concentration to reach conditions where trigger factor is a dimer. Does this influence the results?

To address this, GdnHCl-denatured GAPDH was also diluted 50-fold into refolding buffer resulting in a much lower final concentration of GdnHCl of 0.06 M. We then measured the recovery of GAPDH activity in the presence of TF. Our activity assay showed that trigger factor behaves in a very similar concentration-dependent manner independent of the residual amount of GdnHCl (0.26 M GdnHCl or 0.06 M GdnHCl) (Supplementary Fig. 3b-c). Note that the K_d for TF dimerization measured in 0.06 M GdnHCl buffer at $29 \pm 4 \mu\text{M}$ is very close to the one measured in native buffer that contains no denaturant ($18.4 \pm 0.8 \mu\text{M}$) and is much lower than the one measured in 0.26 M GdnHCl buffer at $76 \pm 6 \mu\text{M}$. (Supplementary Fig. 3a). In addition, we prepared urea-denatured GAPDH and used this in our refolding assay (the final concentration of urea was 0.16 M urea or 0.7 M urea) (Supplementary Fig. 3d-e). Trigger factor also showed a fairly similar behavior in GAPDH refolding assays, suggesting that trigger factor-mediated GAPDH refolding is rather independent of residual denaturant. One reason for this is that the K_d of trigger factor binding to refolding GAPDH ($\sim 0.3 \mu\text{M}$) is more than 50 fold lower than the K_d for TF dimerization ($\sim 18 \mu\text{M}$). We reasoned that the presence of client should readily trigger TF monomerization, allowing monomeric trigger factor to associate with its clients. This suggests the dimerization of TF has minimal effect on client binding or folding. All these new data are included in supplementary figure 3.

Are differences observed with time in the HDX pattern of GAPDH when in complex with trigger factor? It will be interesting to see if GAPDH undergoes structural fluctuations while bound to trigger factor. The overall structure of GAPDH is shown to be dynamic in complex with the trigger factor. It is stated that trigger factor primarily binds to the oligomerization interface and aggregation prone regions of GAPDH. These regions are shown to undergo maximum deuterium uptake in HDX and are most dynamic in complex with trigger factor. Shouldn't they be more protected from exchange if they are tightly bound to trigger factor?

We agree that it would be interesting to see if GAPDH undergoes structural fluctuations while bound to trigger factor. However, in order to address this question we would need to perform pulsed HDX experiments. Such an analysis is beyond the scope of the current manuscript.

With regards to the reviewer's second point, our analysis is confounded by the fact that we are performing comparative HDX and comparing GAPDH under refolding conditions, where GAPDH oligomerizes (Fig 1) to TF-bound GAPDH. GAPDH will be in a different state in each of these conditions. In the samples containing only GAPDH, oligomers form whereas GAPDH binds to TF as a monomer. This means that in GAPDH alone, the oligomeric interfaces will be protected from exchange, whereas when bound to TF residues involved in TF-GAPDH contacts will be protected (and not those in oligomeric interfaces). Thus the protection expected in GAPDH upon binding TF is likely lost in

larger structural changes due to differences in the oligomeric state of the protein. We have clarified this on page 11 of the revised manuscript.

How is GAPDH released from trigger factor? For the enzymatic activity, very high dilutions may trigger release of GAPDH. Does aggregation occur under these conditions upon spontaneous refolding? Are such high dilutions physiologically relevant or are these studies meant to demonstrate basic properties of the complex?

We do not know exactly how clients are released from trigger factor in vivo. The reviewer is right that the high dilution that we used may trigger client release under our in vitro conditions. We have not measured aggregation under these conditions in vitro perhaps due to the low concentration (sub-nanomolar) after as aggregation is a higher order reaction and thus very sensitive to concentrations. For stress related chaperones like Spy dilution may play a real physiological role in client release as the concentration of the chaperone declines precipitously during stress recovery. However for a non-stress activated chaperone like trigger factor, whose concentration is high and constitutive dilutions are unlikely to be physiologically relevant but do provide a convenient way to trigger release and are necessary for the activity assay.

The characterization of GAPDH bound to trigger factor by fluorescence provides interesting insight into the folding state. As this is one of the core findings of the paper additional results to support and extent the conclusion that it is a near native state would further strengthen the study.

In the revised manuscript, we employed additional experiments, including circular dichroism and limited proteolysis to obtain more evidence showing that GAPDH folds into a near-native state while bound to trigger factor. These new results are shown in Fig. 3. In short, we showed that GAPDH regains secondary structure and forms a more protease-resistant state while bound to trigger factor. Note that our evidence indicated that this folding state is close to but not identical to the native state, thus we called this folding state as a near-native state.

For these experiments a trigger factor variant lacking the intrinsic Trp residue is used. I am sure the authors have performed experiments shown for wt trigger factor also with this variant. These should be included in the supplement.

As the reviewer suggested, we performed the GAPDH activity assay using WT trigger factor and W151F trigger factor. The results showed very similar foldase activities for WT trigger factor and W151F variant, suggesting that the substitution of this poorly conserved tryptophane with phenylalanine (a substitution that also occurs in evolution, see the sequence alignment below) does not substantially affect its activity.

Gammaproteobacteria	DAIEVEKPTCEVNDADVDMAMLETLRKQHA	KAVDREAGDNDRAKIDFTGSIDGEEFEGG	180
Ferrimonas.senticii	DAIEVEKPTCEVSDADLDNMLETLRKQHAF	NAVEREAGDNDRVKLNFGVSGIDGEEFEGG	180
Paraferrimonas.sedimenticola	AAIEVEKPTCEVKDEDVDAMIETLRKQHATY	KSVKRKAKKGDVKLNFGVSGIDGEEFEGG	179
Shewanella.sp.Scap07	DTIEVEQPTAEVTEADVDMAMLETLRKQHAT	FEAVEREAAEGDKAKINFGVSDGEEFEGG	179
Parashewanella.spongiae	DAIEVEKPTSDVTDADVDSMIETLRKQHAT	FEVVERAVEDGDKVMNFVSGVSDGEEFEGG	179
Tatumella.citrea	ETIEVEKPVAEVKDEDVDAMLDTLRKQQAD	WKETDAAATAEDRVTIDFGVSGVSDGEEFEGG	180
Enterobacteriales	ESIEVEKPVVEVTDADVDMAMLDTLRKQQAT	WATERAVEAEDRVTIDFGVSGVSDGEEFEGG	180
Shimwellia.pseudoproteus	DAIEVEKPVVEVTDADVDTMLDLTKRQQAT	WTKDGAADAEDRVTIDFGVSGVSDGEEFEGG	180
Yersiniaceae	ESIEVEKPVVEVNDADVDTMLETLRKQQAT	WKETDAAATAEDRATLDFVSGVSDGEEFEGG	180
Pasteurellaceae	ENIKVEKPVTEISDADIDKMIETLRKQQAT	FAETAEAAKADDRVTIDFGVSGVSDGEEFEGG	180
Haemophilus.parasuis	ENIEVEKPVVEITEADLDMVDVLRKQQAT	FAETTEAAKADDRVTIDFGVSGVSDGEEFEGG	180
Nicotellia.semolina	ENIEVEKPVVDIAEADLDMIDVLRKQQAT	FAESQEAAKADDRVTIDFGVSGVSDGEEFEGG	180

::* . : : *:* *:::*:* * . *:::* **:* **

An effort should be made to integrate the results of previous studies on trigger factor and GAPDH (Huang et al.; Liu et al.) in more detail into the discussion of the current results.

We thank the reviewer for this suggestion. More in depth discussion has been added in our revised manuscript.

Minor points

Error bars are missing from many graphs.

We apologise for this omission. Error bars are shown in the anisotropy titration and experimental replicates are shown in the kinetic plots to show the reproducibility of experiments.

What is the y-axis shown in Fig. 1c?

The y-axis shown in Fig. 1c (old manuscript) is normalized fluorescence from microscale thermophoresis (MST). In the revised manuscript, we have replaced the MST data with those from anisotropy experiments (Fig.1d) for a couple of reasons.

In the MST experiments, there is no temperature control system, so all the MST experiments were conducted at room temperature, where the refolding GAPDH could aggregate. Thus, the affinity measurements made are incompatible with the conditions used in other experiments in our study. Anisotropy titration experiments, on the other hand, were conducted at 4C, which is closer to the conditions of the other experiments in this study. In addition, the MST experiments appeared to generate much noise data for our systems than anisotropy titration experiment does (see data below). For these reasons, we decided to replaced the MST data with anisotropy titration experiment for the Kd measurement.

MST data	Anisotropy titration
----------	----------------------

In Fig. 1d, extrapolation to zero time point should be shown in a zoomed view. The corresponding figure in the text is mentioned as Fig. 1e when it should be Fig. 1d.

As review suggested, we added a plot of initial anisotropy vs TF concentration in Supplementary Fig. 4b.

Fig. 2c should show a control of native GAPDH activity at similar concentrations used for refolded GAPDH.

The native GAPDH trace has now been included in the plot (Fig. 2b).

Why does the increase in absorbance in Fig. 2c not saturate?

This is simply because these traces were not recorded over a long enough time window to show saturation. Usually, it takes about 2000-25000 sec to see the saturation on these traces under our experimental conditions. To monitor the GAPDH refolding at a given time point, an aliquot was withdrawn from the refolding solution every 30 min. The GAPDH activity was monitored by measuring the absorbance at 340nm. Thus, in these kinetic experiments, the maximum time window for each measurement is ~28 min so that the sample for the next time point can be ready for the measurement..

In Fig. 2d. a control of GAPDH alone and a native mix of GAPDH and trigger factor should be shown.

The main point of this experiment was to show that 1:1 complex is very stable and native GAPDH is not formed during the incubation. After carefully consideration, we think this figure is no longer important and we have removed it in part because our AUC

data already made these points. All AUC samples were cooled down to 4C in the centrifuge over a period of 2-3 hours, until the temperature equilibrated at 4C. Thus, our AUC data already showed that 1:1 TF-GAPDH complex is formed and stable for hours at 4C. We have added a clarifying statement about this in our main text.

The fluorescence values on y-axis for Fig. 3a do not seem to be absolute values as shown in Fig. 3d.

Fig.3a was the results of stopped-flow fluorescence and Fig. 3d was the results of fluorometer. Both are arbitrary units. To directly compare the Trp fluorescence spectrum with kinetics, we decided to monitor kinetics in a fluorometer (Fig.3b in the revised manuscript). The old stopped-flow kinetics data have been moved to the supplementary figure 7.

In the legend to Fig. 5, authors should mention the time point for which the HDX data is shown.

We have amended the legend to Fig. 5 accordingly.

REVIEWERS' COMMENTS

Reviewer #2 (Remarks to the Author):

I have read the revised manuscript and find it to be much improved.

The new figure 3 is very nice and is overall convincing. Fig 3d, arrows should be added to indicate who is TG and GAPDH although this can be inferred from the next figure.

Figure 6 is well addressing the effect of TG on the refolding of another model substrate, *Vibrio harveyi* luciferase, which is moreover from a prokaryote, as in the case of TG.

The role of DnaKJE was well addressed in Supplementary Fig. 5

New figure A is convincing.

Given that TG does not seem to undergo multiple turnovers in refolding GAPDH, the term “holding” chaperone was well chosen to replace across the paper the term “holdase” that would have implied a catalytic process.

Consequently, my initial concerns have been convincingly addressed. It appears that this is also the case for the concerns of the other referees.

Reviewer #3 (Remarks to the Author):

The authors have addressed my queries in a satisfactory manner.

The new data clarify open issues.

Reviewer #2 (Remarks to the Author):

I have read the revised manuscript and find it to be much improved.

→ We thank the reviewer for the positive comments

The new figure 3 is very nice and is overall convincing. Fig 3d, arrows should be added to indicate who is TG and GAPDH although this can be inferred from the next figure.

→ As reviewer #2 suggested, we have marked the bands corresponding to TF and GAPDH in Fig 3d.

Figure 6 is well addressing the effect of TG on the refolding of another model substrate, *Vibrio harveyi* luciferase, which is moreover from a prokaryote, as in the case of TG.

The role of DnaKJE was well addressed in Supplementary Fig. 5

New figure A is convincing.

Given that TG does not seem to undergo multiple turnovers in refolding GAPDH, the term “holding” chaperone was well chosen to replace across the paper the term “holdase” that would have implied a catalytic process.

Consequently, my initial concerns have been convincingly addressed. It appears that this is also the case for the concerns of the other referees.

Reviewer #3 (Remarks to the Author):

The authors have addressed my queries in a satisfactory manner.

The new data clarify open issues.

→ We thank the reviewer for the positive comments